# GraphLand: Evaluating Graph Machine Learning Models on Diverse Industrial Data

**Gleb Bazhenov**[*]
HSE University, Yandex Research
gv-bazhenov@yandex-team.ru

**Oleg Platonov**[*]
HSE University, Yandex Research
olegplatonov@yandex-team.ru

**Liudmila Prokhorenkova**
Yandex Research
ostroumova-la@yandex-team.ru

## Abstract

Although data that can be naturally represented as graphs is widespread in real-world applications across diverse industries, popular graph ML benchmarks for node property prediction only cover a surprisingly narrow set of data domains, and graph neural networks (GNNs) are often evaluated on just a few academic citation networks. This issue is particularly pressing in light of the recent growing interest in designing graph foundation models. These models are supposed to be able to transfer to diverse graph datasets from different domains, and yet the proposed graph foundation models are often evaluated on a very limited set of datasets from narrow applications. To alleviate this issue, we introduce GraphLand: a benchmark of 14 diverse graph datasets for node property prediction from a range of different industrial applications. GraphLand allows evaluating graph ML models on a wide range of graphs with diverse sizes, structural characteristics, and feature sets, all in a unified setting. Further, GraphLand allows investigating such previously underexplored research questions as how realistic temporal distributional shifts under transductive and inductive settings influence graph ML model performance. To mimic realistic industrial settings, we use GraphLand to compare GNNs with gradient-boosted decision trees (GBDT) models that are popular in industrial applications and show that GBDTs provided with additional graph-based input features can sometimes be very strong baselines. Further, we evaluate current general-purpose graph foundation models and find that they fail to produce competitive results on our proposed datasets. Our source code and datasets are available at https://github.com/yandex-research/graphland.

## 1 Introduction

Recently, there has been a significant push for data-centric approaches in machine learning. In particular, high-quality, realistic, reliable, and diverse benchmarks are paramount for proper evaluation of the performance of machine learning methods. In the field of graph machine learning (GML), there has recently been a lot of criticism of existing popular benchmark datasets concerning such aspects as lacking practical relevance (Bechler-Speicher et al., 2025), low structural diversity that leaves most of the possible graph structure space not represented (Palowitch et al., 2022; Maekawa et al., 2022), low application domain diversity (Bechler-Speicher et al., 2025), graph structure not being beneficial for the considered tasks (Errica et al., 2020; Li et al., 2024b; Coupette et al., 2025; Bechler-Speicher et al., 2025), potential bugs in the data collection processes leading to incorrect

---

[*]Equal contribution

39th Conference on Neural Information Processing Systems (NeurIPS 2025) Track on Datasets and Benchmarks.

labels (Li et al., 2023) and duplicated graph nodes (Platonov et al., 2023b). While there have recently been efforts to create more realistic graph benchmarks, they focus on more specific domains (e.g., 3D molecular data) that require specialized models. At the same time, benchmarks for the standard and most widespread GML setting of node property prediction in a single large graph have received considerably less attention, and evaluation of the performance of classic graph neural networks and recent graph foundation models is still often limited to a few academic citation networks despite this setting and models developed for it having vast real-world applications in diverse industries.

We believe the historical focus of GNN evaluation on academic citation networks, which represent only a single (and a rather narrow) application domain, is primarily a consequence of the availability of open data of this type, rather than its relevance to real-world applications. At the same time, some of the most classical and simultaneously practically important examples of real-world graphs — social networks, web graphs, and road networks — are surprisingly rarely used for GNN evaluation, perhaps due to the lack of easily accessible high-quality datasets.

Recently, there has been a lot of interest in developing graph foundation models (GFMs) — models that after large-scale pretraining can be applied to diverse graph datasets without or with minimal fine-tuning (Wang et al., 2025; Mao et al., 2024). Proper evaluation of such models thus requires the use of a diverse set of realistic graph datasets. However, currently the proposed GFMs are frequently evaluated only on text-attributed graphs (and mostly citation networks), thus overlooking the problem of transferring to graphs with different node feature sets, which is required for truly general GFMs as graphs in real-world applications from different domains often come with completely different node feature sets.

It has been argued that due to the unavailability of diverse realistic industrial datasets for researchers it is worth shifting the evaluation of GML models to synthetic datasets (Palowitch et al., 2022; Yoon et al., 2023). However, we believe that it is important to evaluate models on real-world data as much as possible, both to obtain unbiased estimates of model performance in realistic scenarios and to showcase the potential of GML in industrial applications. Thus, it is desirable to have open and easily accessible diverse and realistic graph datasets.

In our work, we aim to alleviate the issue of a lack of realistic GNN benchmarks for node property prediction by introducing GraphLand: a collection of graphs and associated machine learning tasks collected from a variety of industrial applications that represent real-world GML usage. GraphLand significantly extends the set of available datasets for GML model evaluation, providing in a unified format 14 graph datasets, many of which represent applications or structural properties that have not been covered by standard GML benchmarks before. The datasets in GraphLand have been collected both from open data that has been underutilized or not utilized at all in the field of GML, and from newly released data from services of a large technological company for which the use of GML has internally proven its usefulness. A key feature of GraphLand is its diversity, with graphs spanning a wide range of domains, sizes, and structural properties, and having rich node features with different types, meanings, and distributions.

For datasets in GraphLand, we provide several data splits, including a realistic temporal one, which allows for investigating practically important questions previously underexplored in GML literature: how temporal distributional shifts affect the performance of GML models in both transductive and inductive settings.

We run extensive experiments on GraphLand datasets with a range of GNNs and current GFMs, as well as with classic gradient-boosted decision trees (GBDT) models (Friedman, 2001) which are popular in industrial applications and which we adapt to graph-structured data by providing them with additional graph-based input features. We find that GNNs can achieve great results in industrial applications with attention-based GNNs often performing better than more classic ones. However, their performance can be strongly affected by temporal distributional shifts and dynamically evolving graph structure, which highlights the importance of developing models more resilient to such changes. Further, we find that current GFMs perform poorly on our datasets and fail to achieve results competitive with more classic methods.

We hope that GraphLand will allow more diverse and realistic evaluation of GML models, as well as encourage research into currently underexplored directions such as designing GML models that are more resilient to temporal distributional shifts and dynamically evolving graphs, and designing GFMs that are truly generalizable to graph data from different domains with different node feature sets.

## 2  Limitations of popular graph machine learning benchmarks

By far the most popular datasets used in modern GML literature are the three academic citation networks `cora`, `citeseer`, and `pubmed` (Giles et al., 1998; McCallum et al., 2000; Sen et al., 2008; Namata et al., 2012; Yang et al., 2016). These datasets became so widespread perhaps because they were used by the foundational work on modern GNNs by Kipf & Welling (2017). However, these datasets only cover a single and rather narrow application of paper subject prediction in citation networks. Another popular set of datasets for GML was introduced by Shchur et al. (2018) and includes academic coauthorship networks `coauthor-cs` and `coauthor-physics`, and e-commerce co-purchasing networks `amazon-computers` and `amazon-photo`. However, all the aforementioned datasets together only cover three applications, while GML methods can be used in a much wider variety of settings. Later, larger-scale graph datasets were introduced in the Open Graph Benchmark (OGB) (Hu et al., 2020). However, out of the five node property prediction datasets, three are academic citation networks (`ogbn-arxiv`, `ogbn-mag`, `ogbn-papers100M`) and one more is an e-commerce co-purchasing network (`ogbn-products`). Thus, OGB does not significantly expand the range of real-world applications available for evaluating GML models. Further, all the aforementioned datasets only provide textual descriptions as node features. However, co-purchasing networks, which represent a very practically important application of GML, in realistic settings come with rich product metadata that can be represented as numerical and categorical features. Popular GML benchmarks currently lack datasets with such metadata. Further, all the aforementioned datasets are *homophilous*, i.e., edges in them typically connect nodes of the same class. It has been shown by Huang et al. (2021) that even very simple models can provide strong results in homophilous networks. Thus, it is important for standard GML benchmarks to also include a wide selection of *non-homophilous* graphs, i.e., graphs in which edges do not have the tendency to connect nodes of the same class. For a long time, the only popular source of non-homophilous graph datasets was the benchmark from Pei et al. (2020). However, it was recently shown by Platonov et al. (2023b) that these datasets have numerous problems including duplicated nodes, small size (leading to noisy evaluation metric estimates), and insufficient class representation (such as the `texas` dataset having a class that consists of only a single node). While Platonov et al. (2023b) introduced several new non-homophilous graph datasets, they were meant to be used to reliably reevaluate the performance of different models in the absence of homophily, rather than represent realistic GML applications. Thus, some of these datasets are synthetic (`minesweeper`), semi-synthetic (`roman-empire`), or have limited node features despite the original data source potentially providing more information about the nodes (`tolokers`, `questions`).

Overall, the currently popular graph datasets for node property prediction do not allow evaluating GNNs and other GML models on a wide range of practically impactful industrial applications.

**Text-attributed graphs and generalization of graph foundation models**  Most of the datasets frequently used for node property prediction only have textual descriptions as node attributes. However, graphs representing real-world networks often have rich and diverse node attributes that go beyond texts and encompass a variety of numerical and categorical features with different meanings and distributions. There has recently been a lot of interest in developing general-purpose GFMs that are expected to generalize to graphs from different domains (Wang et al., 2025; Mao et al., 2024). A key challenge for such GFMs is being able to adapt to graphs with different node feature sets, which is required for a model truly generalizable to different domains. Yet, the GFMs that have been proposed in the current literature typically overlook this challenge and are often only evaluated on text-attributed graphs (Liu et al., 2024; Li et al., 2024a; He et al., 2025). Textual attributes can be easily projected to a common latent feature space by applying pretrained text encoders based on Large Language Models, thus allowing a single GFM to work with different text-attributed graphs. The prevalence of such text-attributed graphs in GML benchmarks has led to most of current GFM research overlooking the problem of generalization to different node feature sets, since it is not required to solve tasks from standard benchmarks. However, this problem is very important for real-world industrial applications of GML in which graphs often come attributed with a mixture of numerical and categorical node features. GFMs must therefore be able to work with such features to effectively solve practical tasks. The problem of devising a single foundation model that can work with arbitrary numerical and categorical features has received significant attention from the ML for tabular data community, where such features are standard, and first successful attempts to develop such a model have recently emerged (Hollmann et al., 2023, 2025; Ye et al., 2023, 2025;

Mueller et al., 2025; Ma et al., 2024; Qu et al., 2025). However, these ideas have not yet spread to the GML community (likely because current standard GML benchmarks do not require working with non-textual node features) despite the significant benefits of designing a successful GFM that can handle arbitrary node feature sets for practical applications.

## 3 GraphLand: a collection of diverse industrial graph datasets

GraphLand is a collection of 14 graph datasets with node property prediction tasks (either classification or regression). Some of these datasets are newly released for this benchmark, while others are collected from open data sources that are underutilized or not utilized at all in current GML benchmarking. In selecting datasets for GraphLand, we aim to fulfill the following desiderata:

- Datasets should come from a range of impactful industrial applications, including such archetypal examples of real-world networks as social networks, web graphs, and road networks;

- Datasets should have graph structure that is beneficial for the considered task;

- Graphs from different datasets should have diverse structural properties;

- Datasets should have rich node attributes consisting of a range of numerical and/or categorical features;

- If a graph is dynamically evolving over time, it should ideally have the necessary temporal information to construct a realistic time-based data split and a meaningful inductive setting;

- Datasets should cover a range of graph sizes to allow for evaluation with different available computational resources, but should contain at least 10,000 nodes to avoid particularly noisy evaluation metric estimates.

In this section, we briefly describe our datasets and the corresponding prediction tasks. More detailed descriptions are provided in Appendix A.1, while the discussion of structural properties and other characteristics of our datasets can be found in Appendix A.2.

First, we describe datasets based on the data newly released specifically for our benchmark.

**web-fraud**, **web-topics**, and **web-traffic** These three datasets represent a part of the Internet (web graph). The nodes are websites, and a directed edge connects two nodes if at least one user followed a link from one website to the other in a selected period of time. We prepared three datasets with the same graph but different tasks: in web-fraud, the task is to predict which websites are fraudulent (strongly imbalanced binary classification); in web-topics, the task is to predict the topic that a website belongs to (multiclass classification); and in web-traffic, the task is to predict how many users visited a website in a specific period of time (regression). With almost 3 million nodes, this is one of the largest publicly available attributed graphs that is not a citation network.

**artnet-views** and **artnet-exp** These two datasets represent a social network of art creators. The nodes are users, and an edge connects two nodes if the users are friends. We prepared two datasets with the same graph but different tasks: in artnet-views, the task is to predict how many views a user receives in a specific period of time (regression); and in artnet-exp, the task is to predict which users tend to create explicit art content (binary classification).

**city-roads-M** and **city-roads-L** These datasets represent road networks of two major cities, with the second one being several times larger than the first. The nodes are segments of roads, and a directed edge connects two nodes if the segments are incident to each other and moving from one segment to the other is permitted by traffic rules. The task is to predict the average traveling speed on the road segment at a specific timestamp (regression).

**city-reviews** This dataset represents a review service of places and organizations in two major cities. The nodes are users who leave ratings and post comments about various places, and an edge connects two nodes if the users often leave reviews for the same organizations. The task is fraud detection — to predict which users leave fraudulent reviews (binary classification).

Further, we find that beyond academic citation networks, there are quite a few sources of open network data available that are, however, rarely or even never used for GML model evaluation. Below, we describe the datasets we obtained from these open sources. For these datasets, we defined

Table 1: Characteristics of the proposed GraphLand datasets.

| | node classification | | | | | | | node regression | | | | | | |
|---|---|---|---|---|---|---|---|---|---|---|---|---|---|---|
| | hm-categories | pokec-regions | web-topics | tolokers-2 | city-reviews | artnet-exp | web-fraud | hm-prices | avazu-ctr | city-roads-M | city-roads-L | twitch-views | artnet-views | web-traffic |
| # nodes | 46.5K | 1.6M | 2.9M | 11.8K | 148.8K | 50.4K | 2.9M | 46.5K | 76.3K | 57.1K | 142.3K | 168.1K | 50.4K | 2.9M |
| # edges | 10.7M | 22.3M | 12.4M | 519.0K | 1.2M | 280.3K | 12.4M | 10.7M | 11.0M | 107.1K | 231.6K | 6.8M | 280.3K | 12.4M |
| avg degree | 460.92 | 27.32 | 8.56 | 88.28 | 15.66 | 11.12 | 8.56 | 460.92 | 288.04 | 3.75 | 3.26 | 80.87 | 11.12 | 8.56 |
| median degree | 45 | 13 | 2 | 30 | 4 | 2 | 2 | 45 | 71 | 4 | 3 | 32 | 2 | 2 |
| avg distance | 2.45 | 4.68 | 3.08 | 2.79 | 4.91 | 4.42 | 3.08 | 2.45 | 3.55 | 126.75 | 194.05 | 2.88 | 4.42 | 3.08 |
| diameter | 13 | 14 | 36 | 11 | 19 | 13 | 36 | 13 | 14 | 383 | 553 | 8 | 13 | 36 |
| global clustering | 0.27 | 0.05 | 0.00 | 0.23 | 0.26 | 0.03 | 0.00 | 0.27 | 0.24 | 0.00 | 0.00 | 0.02 | 0.03 | 0.00 |
| avg local clustering | 0.70 | 0.11 | 0.33 | 0.53 | 0.41 | 0.08 | 0.33 | 0.70 | 0.85 | 0.00 | 0.00 | 0.16 | 0.08 | 0.33 |
| degree assortativity | $-0.35$ | 0.00 | $-0.14$ | $-0.08$ | 0.01 | 0.03 | $-0.14$ | $-0.35$ | $-0.30$ | 0.70 | 0.74 | $-0.09$ | 0.03 | $-0.14$ |
| # classes | 21 | 183 | 28 | 2 | 2 | 2 | 2 | N/A | N/A | N/A | N/A | N/A | N/A | N/A |
| unbiased homophily | 0.38 | 0.98 | 0.55 | 0.10 | 0.69 | 0.28 | 0.32 | N/A | N/A | N/A | N/A | N/A | N/A | N/A |
| target assortativity | N/A | N/A | N/A | N/A | N/A | N/A | N/A | 0.12 | 0.18 | 0.74 | 0.72 | $-0.41$ | 0.19 | $-0.21$ |
| # node features | 35 | 56 | 263 | 16 | 37 | 75 | 266 | 41 | 260 | 26 | 26 | 4 | 50 | 267 |

or extended node features and labels, as well as constructed the graph structure where it was not explicitly available.

**avazu-ctr** This dataset is based on the data about user interactions with ads provided by the Avazu company (Wang & Cukierski, 2014). The nodes are devices used to access the Internet, and an edge connects two nodes if the devices often visit the same websites. The task is to predict advertisement click-through rate for devices (regression).

**hm-categories** and **hm-prices** These two datasets are based on the product co-purchasing network from the H&M company (García Ling et al., 2022). The nodes are products, and an edge connects two nodes if the products are often bought by the same customers. We prepared two datasets with the same graph but different tasks: in hm-categories, the task is to predict the product category (multiclass classification); and in hm-prices, the task is to predict the product price (regression).

**pokec-regions** This dataset is based on the data from Takac & Zabovsky (2012). It represents the online social network Pokec. The nodes are users, and a directed edge connects two nodes if one user has marked the other as a friend. The task is to predict which region a user is from (extreme multiclass classification with 183 classes).

**twitch-views** This dataset is based on the data from Rozemberczki & Sarkar (2021). It represents the live-streaming network Twitch. The nodes are users, and an edge connects two nodes if both users follow each other. The task is to predict how many views a user receives in a specific period of time (regression).

**tolokers-2** This is a new version of the dataset tolokers from Platonov et al. (2023b); Likhobaba et al. (2023) with a significantly extended set of node features. This dataset represents a network of tolokers (workers) from the Toloka crowdsourcing platform. The nodes are tolokers and an edge connects two nodes if the tolokers have worked on the same task. The task is fraud detection — to predict which tolokers have been banned in one of the projects (binary classification).

Overall, our datasets cover a diverse range of industrial applications. In Table 1, we provide some characteristics of the datasets: as can be seen, our datasets also exhibit very diverse graph structural properties. A more detailed discussion of these characteristics is provided in Appendix A.2.

# 4 Data splits and experimental settings

In GML literature, there are two most popular settings for node property prediction regarding the relative sizes of train, validation, and test sets: one with a high label rate and one with a low label rate. In the high label rate setting, the train set encompasses 50% of all graph nodes or more. This setting is common in heterophilous benchmarks, e.g., it is used by the datasets from Pei et al. (2020) and Platonov et al. (2023b), as well as by most datasets from OGB (Hu et al., 2020). In the low label rate setting, much smaller train set sizes are used (typically, no more than 10% of all graph nodes). This setting is commonly used with the classic cora, citeseer, pubmed citation networks, and it is also used by the ogbn-products dataset from OGB. Both of these settings can appear in real-world

GML usage scenarios, depending on the resources available for data labeling. It is important to provide predetermined splits to ensure experiments in different works are run in the same setting and their results are comparable, but it is also important to accommodate different needs of different research projects. Thus, for datasets in our benchmark we provide fixed splits for both settings. We refer to these splits as the `RL` (random low) and `RH` (random high) splits. Specifically, the `RL` split randomly divides nodes into train/validation/test sets with $10\%/10\%/80\%$ proportions, while the `RH` split randomly divides nodes into train/validation/test sets with $50\%/25\%/25\%$ proportions.

The `RL` and `RH` data splits are random, as is common in current GML benchmarks. However, in real-world applications, data splits are often *temporal*, i.e., the labeled objects are the ones that appeared in the network earlier, while those that appeared later are not labeled and belong to the test set. Despite their prevalence in applications, temporal splits are very rarely used in current GML node property prediction benchmarks. To the best of our knowledge, the only such datasets with temporal splits available are citation networks from OGB, which represent only a single application. At the same time, temporal data splits may significantly affect the prediction problem and the model performance, as they often result in *distributional shifts* between train, validation, and test data. While some types of distributional shifts have been previously explored in the GML literature, e.g., shifts in node features (Gui et al., 2022) and shifts in graph structure (Bazhenov et al., 2023), realistic temporal distributional shifts often combine shifts in several aspects of data simultaneously (e.g., shifts in the distributions of node features, labels, and graph structural characteristics), and the effect of such realistic shifts on GML model performance is currently under-explored. To close this gap, we provide a temporal split for most datasets in our benchmark. We refer to this split as the `TH` (temporal high) split; it divides nodes into train/validation/test sets with $50\%/25\%/25\%$ proportions, i.e., exactly the same proportions as in the `RH` split, which allows comparing model results between the `RH` and `TH` splits to see how the complexity of the task changes when temporal distributional shifts are introduced.

Further, many real-world networks are not static, but evolve over time. Thus, in many applications, not only are there no labels available for nodes that appear in the network later, but the nodes themselves (with their attributes and incident edges) are not available at training time. This setting is known in GML as the *inductive* setting. In contrast to the *transductive* setting in which the whole graph is available at training time (including the nodes for which predictions should be made), in the inductive setting validation and test nodes are not available at training time. Despite temporally evolving graphs being common in practical applications, most standard node property prediction datasets only provide the transductive setting. While it is well-known that GNNs, in contrast to some other GML methods like shallow node embeddings, can work not only in the transductive but also in the inductive setting (Hamilton et al., 2017), it is not well-explored how the lack of complete graph information at training time in the inductive setting affects GNN performance. Moreover, when the inductive setting is used for GNN evaluation in the current literature, the validation and test nodes are typically chosen randomly, which is not realistic, since in real-world applications the inductive setting is almost always induced by the temporal evolution of the graph. To fill this gap, for all datasets in our benchmark for which temporal information is available, we additionally provide the inductive experimental setting. We refer to this setting as `THI` (temporal high / inductive); it has the exact same data split as the transductive `TH` setting, but provides three snapshots of the graph: one for training, one for validation, and one for testing. This allows investigating how model performance changes between the transductive and the inductive setting. To the best of our knowledge, our work is the first to compare GNN performance between random and temporal data splits in the transductive setting under the same split ratios, and between transductive and inductive settings under the same temporal data split. This comparison allows us to investigate how much these differences can affect GNN performance.

## 5 Experiments

### 5.1 Models

In our experiments, we use a range of models which we describe in this subsection. For all models, we conduct extensive hyperparameter tuning. The details of it, as well as the description of other aspects of our experimental setting and more detailed model descriptions, are provided in Appendix B.

**Graph-agnostic baselines** The tasks that we consider in our benchmark, such as fraud detection and CTR prediction, are frequently tackled in industrial settings without considering the graph structure. Thus, as baselines, we use several models that do not have access to the graph structure. We call such models *graph-agnostic*. Comparing results between graph-agnostic and graph-aware models might reveal how much benefit using the graph structure brings for the considered task. First, as a simple baseline, we use ResMLP — an MLP augmented with skip-connections (He et al., 2016) and layer normalization (Ba et al., 2016). This model treats all nodes as independent data samples and does not use the graph structure. It has been shown that models of this type can serve as very strong baselines for industrial data with mixed numerical and categorical features (Gorishniy et al., 2021). Further, we consider GBDT models (Friedman, 2001). These models are very popular in industrial applications and are often considered to be better than neural networks at dealing with numerical features (Gorishniy et al., 2021, 2022) and regression tasks (Zhang et al., 2023), which makes them very relevant baselines for our benchmark. We use the three most popular GBDT implementations: XGBoost (Chen & Guestrin, 2016), LightGBM (Ke et al., 2017), and CatBoost (Prokhorenkova et al., 2018).

Further, we attempt to make originally graph-agnostic models stronger on our benchmark by providing them with some graph information through feature augmentation. Specifically, for each node in the graph, we aggregate features from all its one-hop neighbors in the graph, compute the mean, maximum, and minimum values of each feature, and append these statistics to the node's original features. This mimics one step of GNN spatial graph convolution and allows graph-agnostic models to use some of the graph information that GNNs use. We call this feature augmentation strategy *neighborhood feature aggregation* (NFA) and denote models that use it with the -NFA suffix. A more detailed description of NFA is provided in Appendix B.2.

**Graph neural networks** As our main models, we use four representative GNN architectures: two classic GNNs — GCN (Kipf & Welling, 2017) and GraphSAGE (Hamilton et al., 2017), and two attention-augmented GNNs — GAT (Veličković et al., 2018) and neighborhood-attention Graph Transformer (GT) (Shi et al., 2021). Note that GT is a local graph transformer with attention only over node neighborhoods, which is different from global graph transformers with all-to-all attention. For all GNNs, we use the modifications from Platonov et al. (2023b) that add skip connections, layer normalization, and MLP blocks to the models. We found that these modifications often significantly improve performance on our datasets.

It has been argued recently that in some GML datasets the graph structure does not actually help solve the considered tasks (Errica et al., 2020; Li et al., 2024b; Coupette et al., 2025; Bechler-Speicher et al., 2025). Thus, when new graph datasets are introduced, it is important to experimentally verify that their graph structure is actually helpful. For this, the performance of graph-agnostic and graph-aware models can be compared. To avoid other factors influencing the comparison, the two models being compared should be otherwise identical except that one has access to the graph structure and the other does not. Our set of models provides two such comparisons. First, graph-agnostic models can be compared with their NFA-augmented versions, which have access to graph neighborhood feature information. Second, ResMLP can be compared to GNNs: our GNNs have the exact same backbone that our ResMLP baseline has, except GNNs have spatial graph convolution modules added, which makes the comparison fair.

**Graph foundation models** Recently, there has been a lot of interest in developing graph foundation models — models that can be applied to diverse graph datasets without or with minimal fine-tuning. However, we found that the current GFMs predominantly focus on text-attributed graphs and overlook the challenge of transferring a single model to datasets with different node feature sets. This prevents such models from being truly general, as graphs in real-world applications often have many different numerical and/or categorical node attributes that are important for solving the relevant tasks. We have found only a few GFMs that support node property prediction in graphs with arbitrary node attributes: SAMGPT (Yu et al., 2025), GCOPE (Zhao et al., 2024), OpenGraph (Xia et al., 2024), GraphFM (Lachi et al., 2024), AnyGraph (Xia & Huang, 2024), and TS-GNN (Finkelshtein et al., 2025). Of these models, only OpenGraph and AnyGraph have publicly available weights. Thus, we evaluate these two models. We commend the researchers who make their GFMs openly available and encourage others to do the same. For both models, we use the in-context learning (ICL) setting recommended by the authors for node classification, in which the task is cast as predicting links to virtual class nodes. Further, we were able to reproduce the pretraining of TS-GNN and GCOPE, and

Table 2: Experimental results under the `RL` (random low) data split in the transductive setting. The best result and those statistically indistinguishable from it are highlighted in orange. `TLE` stands for time limit exceeded (24 hours); `MLE` stands for memory limit exceeded (80 GB VRAM); `RTE` stands for runtime error in the official code of GFMs.

(a) Results for classification datasets. Accuracy is reported for multiclass classification datasets and Average Precision is reported for binary classification datasets.

| | multiclass classification | | | binary classification | | | |
|---|---|---|---|---|---|---|---|
| | hm-categories | pokec-regions | web-topics | tolokers-2 | city-reviews | artnet-exp | web-fraud |
| best const. pred. | $19.46 \pm 0.00$ | $3.77 \pm 0.00$ | $28.36 \pm 0.00$ | $21.82 \pm 0.00$ | $12.09 \pm 0.00$ | $10.00 \pm 0.00$ | $0.66 \pm 0.00$ |
| ResMLP | $37.72 \pm 0.18$ | $4.88 \pm 0.01$ | $42.41 \pm 0.02$ | $41.16 \pm 1.13$ | $71.32 \pm 0.11$ | $35.07 \pm 2.34$ | $8.77 \pm 0.18$ |
| XGBoost | $40.04 \pm 0.09$ | $4.93 \pm 0.01$ | TLE | $45.76 \pm 1.00$ | $74.70 \pm 0.13$ | $41.92 \pm 0.82$ | $11.54 \pm 0.04$ |
| LightGBM | $39.73 \pm 0.08$ | $4.89 \pm 0.00$ | TLE | $44.60 \pm 0.12$ | $74.51 \pm 0.04$ | $41.21 \pm 0.12$ | TLE |
| CatBoost | $40.72 \pm 0.40$ | TLE | TLE | $46.10 \pm 0.35$ | $74.77 \pm 0.10$ | $42.50 \pm 0.12$ | TLE |
| ResMLP-NFA | $48.72 \pm 0.38$ | $8.05 \pm 0.03$ | MLE | $48.14 \pm 1.40$ | $76.02 \pm 0.14$ | $38.25 \pm 0.56$ | MLE |
| LightGBM-NFA | $56.55 \pm 0.15$ | $9.53 \pm 0.01$ | TLE | $56.16 \pm 0.28$ | $78.33 \pm 0.04$ | $45.40 \pm 0.13$ | TLE |
| GCN | $61.70 \pm 0.35$ | $34.96 \pm 0.38$ | $46.45 \pm 0.10$ | $51.32 \pm 0.96$ | $77.15 \pm 0.28$ | $43.09 \pm 0.38$ | $10.02 \pm 0.18$ |
| GraphSAGE | $56.75 \pm 0.53$ | $37.88 \pm 0.41$ | $47.41 \pm 0.13$ | $53.73 \pm 0.53$ | $77.82 \pm 0.13$ | $42.65 \pm 0.59$ | $12.11 \pm 0.23$ |
| GAT | $67.96 \pm 0.33$ | $46.17 \pm 0.32$ | $48.25 \pm 0.05$ | $53.78 \pm 1.34$ | $77.67 \pm 0.13$ | $46.62 \pm 0.32$ | $13.32 \pm 0.29$ |
| GT | $69.23 \pm 0.50$ | $46.47 \pm 0.16$ | $48.00 \pm 0.05$ | $54.50 \pm 1.20$ | $76.97 \pm 0.21$ | $45.16 \pm 0.46$ | $12.74 \pm 0.42$ |
| OpenGraph (ICL) | $9.49 \pm 0.93$ | $1.73 \pm 0.31$ | RTE | $40.49 \pm 0.31$ | $58.44 \pm 1.08$ | $15.65 \pm 1.23$ | RTE |
| AnyGraph (ICL) | $15.47 \pm 2.36$ | $24.65 \pm 1.51$ | $6.67 \pm 3.88$ | $31.33 \pm 2.89$ | $64.37 \pm 1.29$ | $13.14 \pm 1.15$ | $0.68 \pm 0.03$ |
| TS-GNN (ICL) | $20.09 \pm 1.29$ | MLE | MLE | $38.54 \pm 0.94$ | $43.46 \pm 5.17$ | $20.44 \pm 1.05$ | MLE |
| GCOPE (FT) | $19.51 \pm 0.07$ | TLE | TLE | $28.67 \pm 1.42$ | $67.38 \pm 1.23$ | $16.10 \pm 2.79$ | TLE |

(b) Results for regression datasets. $R^2$ is reported for all datasets.

| | hm-prices | avazu-ctr | city-roads-M | city-roads-L | twitch-views | artnet-views | web-traffic |
|---|---|---|---|---|---|---|---|
| best const. pred. | $0.00 \pm 0.00$ | $0.00 \pm 0.00$ | $0.00 \pm 0.00$ | $0.00 \pm 0.00$ | $0.00 \pm 0.00$ | $0.00 \pm 0.00$ | $0.00 \pm 0.00$ |
| ResMLP | $62.66 \pm 0.37$ | $24.54 \pm 0.36$ | $54.77 \pm 0.15$ | $46.47 \pm 0.29$ | $13.35 \pm 0.02$ | $29.71 \pm 0.60$ | $72.42 \pm 0.05$ |
| XGBoost | $65.68 \pm 0.16$ | $26.72 \pm 0.02$ | $59.14 \pm 0.11$ | $53.75 \pm 0.07$ | $13.39 \pm 0.00$ | $32.74 \pm 0.04$ | TLE |
| LightGBM | $65.44 \pm 0.09$ | $25.83 \pm 0.04$ | $57.76 \pm 0.10$ | $52.65 \pm 0.08$ | $13.38 \pm 0.01$ | $32.47 \pm 0.04$ | TLE |
| CatBoost | $66.85 \pm 0.28$ | $26.10 \pm 0.04$ | $57.53 \pm 0.18$ | $51.43 \pm 0.17$ | $13.20 \pm 0.03$ | $32.89 \pm 0.05$ | TLE |
| ResMLP-NFA | $67.19 \pm 0.30$ | $31.11 \pm 0.30$ | $57.82 \pm 0.14$ | $50.85 \pm 0.18$ | $51.43 \pm 0.60$ | $51.03 \pm 0.41$ | MLE |
| LightGBM-NFA | $70.46 \pm 0.09$ | $31.72 \pm 0.06$ | $61.00 \pm 0.05$ | $55.26 \pm 0.04$ | $60.20 \pm 0.01$ | $56.55 \pm 0.04$ | TLE |
| GCN | $69.76 \pm 0.38$ | $30.47 \pm 0.27$ | $59.05 \pm 0.16$ | $53.26 \pm 0.14$ | $75.55 \pm 0.05$ | $55.99 \pm 0.26$ | $82.07 \pm 0.14$ |
| GraphSAGE | $70.54 \pm 0.21$ | $31.84 \pm 0.24$ | $57.51 \pm 0.53$ | $52.43 \pm 0.25$ | $66.87 \pm 0.11$ | $49.79 \pm 0.51$ | $83.50 \pm 0.13$ |
| GAT | $73.17 \pm 0.50$ | $33.20 \pm 0.20$ | $59.11 \pm 0.20$ | $53.43 \pm 0.20$ | $72.93 \pm 0.17$ | $53.36 \pm 0.78$ | $84.68 \pm 0.06$ |
| GT | $71.87 \pm 0.65$ | $30.87 \pm 0.47$ | $58.05 \pm 0.58$ | $53.38 \pm 0.12$ | $72.19 \pm 0.14$ | $54.23 \pm 0.22$ | $84.49 \pm 0.07$ |

we also evaluate these two models (specifically, for TS-GNN we use the TS-Mean variant). We use the ICL setting for TS-GNN and the fine-tuning (FT) setting for GCOPE, as recommended by the authors (note that TS-GNN solves several least squares problems to fit its weights before making predictions, which can be considered a form of training, but, following the authors of this model, we still count it as ICL). However, neither of the considered models supports node regression; thus, we are only able to evaluate these models on our node classification datasets (while TS-GNN and GCOPE can support node regression in theory, their official implementations do not provide such support). Further, our experiments show that the considered GFMs cannot scale to large datasets.

## 5.2 Low label rate, random data split, transductive setting

The results for experiments under the random low label rate data split (`RL`) are provided in Table 2.

First, we can see that the graph structure in our datasets is very beneficial for the considered tasks as all GNNs always significantly outperform ResMLP and NFA-augmented models always significantly outperform their counterparts without NFA. Providing graph-agnostic models with graph information through NFA significantly improves their performance and even allows LightGBM-NFA to achieve the best results on four datasets including three regression ones. At the same time, GNNs outperform ResMLP-NFA on these datasets, which suggests that the success of LightGBM-NFA is due to GBDTs being better at handling numerical features and regression targets (Gorishniy et al., 2021, 2022; Zhang et al., 2023). This highlights that GBDTs with augmented features are strong baselines in realistic industrial settings. Still, GNNs manage to outperform them on most datasets, showing their effectiveness for industrial applications.

Table 3: Experimental results under the RH (random high), TH (temporal high), and THI (temporal high / inductive) settings. The best result and those statistically indistinguishable from it are highlighted in red for RH, violet for TH, and blue for THI.

(a) Results for classification datasets. Accuracy is reported for multiclass classification datasets and Average Precision is reported for binary classification datasets.

| | | multiclass classification | | | binary classification | | |
|---|---|---|---|---|---|---|---|
| | | hm-categories | pokec-regions | web-topics | tolokers-2 | artnet-exp | web-fraud |
| best const. pred. | RH | $19.46 \pm 0.00$ | $3.77 \pm 0.00$ | $28.36 \pm 0.00$ | $21.82 \pm 0.00$ | $10.00 \pm 0.00$ | $0.66 \pm 0.00$ |
| | TH | $19.57 \pm 0.00$ | $2.98 \pm 0.00$ | $23.28 \pm 0.00$ | $8.61 \pm 0.00$ | $7.84 \pm 0.00$ | $0.15 \pm 0.00$ |
| | THI | $19.57 \pm 0.00$ | $2.98 \pm 0.00$ | $23.28 \pm 0.00$ | $8.61 \pm 0.00$ | $7.84 \pm 0.00$ | $0.15 \pm 0.00$ |
| LightGBM | RH | $47.28 \pm 0.14$ | $5.10 \pm 0.01$ | TLE | $49.92 \pm 0.19$ | $44.98 \pm 0.80$ | TLE |
| | TH | $32.08 \pm 0.25$ | $4.13 \pm 0.01$ | TLE | $15.85 \pm 3.89$ | $37.34 \pm 0.18$ | TLE |
| | THI | $32.08 \pm 0.25$ | $4.13 \pm 0.01$ | TLE | $15.85 \pm 3.89$ | $37.34 \pm 0.18$ | TLE |
| LightGBM-NFA | RH | $67.09 \pm 0.15$ | $12.15 \pm 0.02$ | TLE | $62.19 \pm 0.35$ | $49.26 \pm 0.18$ | TLE |
| | TH | $52.45 \pm 0.18$ | $5.54 \pm 0.01$ | TLE | $31.81 \pm 7.51$ | $39.89 \pm 0.37$ | TLE |
| | THI | $47.46 \pm 0.28$ | $4.58 \pm 0.02$ | TLE | $45.45 \pm 0.82$ | $39.91 \pm 0.25$ | TLE |
| GCN | RH | $73.38 \pm 0.42$ | $35.08 \pm 0.62$ | $48.88 \pm 0.09$ | $60.49 \pm 0.86$ | $49.80 \pm 0.35$ | $15.58 \pm 0.20$ |
| | TH | $56.91 \pm 0.55$ | $11.88 \pm 0.31$ | $38.20 \pm 0.19$ | $46.72 \pm 1.19$ | $40.64 \pm 0.40$ | $4.85 \pm 1.42$ |
| | THI | $47.05 \pm 1.69$ | $6.88 \pm 0.51$ | $37.76 \pm 0.06$ | $32.43 \pm 8.03$ | $41.28 \pm 0.28$ | $3.44 \pm 0.33$ |
| GraphSAGE | RH | $73.34 \pm 0.68$ | $40.76 \pm 0.21$ | $50.05 \pm 0.03$ | $58.42 \pm 0.92$ | $48.49 \pm 0.37$ | $20.47 \pm 0.17$ |
| | TH | $59.62 \pm 0.51$ | $16.60 \pm 0.28$ | $39.00 \pm 0.09$ | $17.05 \pm 7.65$ | $40.50 \pm 0.84$ | $16.01 \pm 2.22$ |
| | THI | $48.11 \pm 2.08$ | $8.04 \pm 0.26$ | $38.04 \pm 0.18$ | $30.86 \pm 9.48$ | $40.53 \pm 0.40$ | $13.88 \pm 1.32$ |
| GAT | RH | $79.19 \pm 0.21$ | $46.72 \pm 0.69$ | $50.54 \pm 0.04$ | $63.76 \pm 1.30$ | $50.62 \pm 0.35$ | $20.43 \pm 0.21$ |
| | TH | $61.28 \pm 0.97$ | $20.43 \pm 0.55$ | $39.24 \pm 0.23$ | $38.59 \pm 6.19$ | $41.85 \pm 0.63$ | $16.50 \pm 1.14$ |
| | THI | $59.34 \pm 1.09$ | $13.38 \pm 0.35$ | $38.77 \pm 0.35$ | $24.53 \pm 9.55$ | $41.64 \pm 0.32$ | $11.98 \pm 1.54$ |
| GT | RH | $79.28 \pm 0.31$ | $50.06 \pm 0.53$ | $50.58 \pm 0.04$ | $60.32 \pm 1.21$ | $49.32 \pm 1.00$ | $19.73 \pm 0.34$ |
| | TH | $63.31 \pm 0.45$ | $25.09 \pm 0.58$ | $39.19 \pm 0.15$ | $34.15 \pm 4.81$ | $40.10 \pm 0.60$ | $11.97 \pm 1.13$ |
| | THI | $59.54 \pm 1.59$ | $17.22 \pm 0.42$ | $38.78 \pm 0.08$ | $22.89 \pm 10.40$ | $40.26 \pm 0.82$ | $7.84 \pm 2.35$ |
| OpenGraph (ICL) | RH | $11.69 \pm 0.84$ | $2.56 \pm 0.42$ | RTE | $44.62 \pm 1.35$ | $23.72 \pm 1.86$ | RTE |
| | TH | $5.76 \pm 1.03$ | $0.80 \pm 0.45$ | RTE | $9.12 \pm 1.74$ | $16.19 \pm 1.36$ | RTE |
| | THI | $5.76 \pm 1.03$ | $0.80 \pm 0.45$ | RTE | $9.12 \pm 1.74$ | $16.19 \pm 1.36$ | RTE |
| AnyGraph (ICL) | RH | $15.65 \pm 2.82$ | $27.67 \pm 2.48$ | $6.30 \pm 2.82$ | $30.21 \pm 3.32$ | $15.80 \pm 1.90$ | $0.67 \pm 0.02$ |
| | TH | $9.47 \pm 1.13$ | $9.20 \pm 0.67$ | $11.14 \pm 5.16$ | $13.52 \pm 4.74$ | $11.80 \pm 1.01$ | $0.16 \pm 0.01$ |
| | THI | $9.47 \pm 1.13$ | $9.20 \pm 0.67$ | $11.14 \pm 5.16$ | $13.52 \pm 4.74$ | $11.80 \pm 1.01$ | $0.16 \pm 0.01$ |
| TS-GNN (ICL) | RH | $15.48 \pm 1.93$ | MLE | MLE | $31.83 \pm 2.55$ | $13.38 \pm 2.59$ | MLE |
| | TH | $18.91 \pm 0.32$ | MLE | MLE | $12.59 \pm 1.97$ | $9.46 \pm 1.25$ | MLE |
| | THI | $18.91 \pm 0.32$ | MLE | MLE | $12.59 \pm 1.97$ | $9.46 \pm 1.25$ | MLE |
| GCOPE (FT) | RH | $19.99 \pm 0.12$ | TLE | TLE | $31.79 \pm 1.95$ | $23.86 \pm 2.33$ | TLE |
| | TH | $19.14 \pm 0.58$ | TLE | TLE | $8.46 \pm 1.15$ | $20.83 \pm 1.18$ | TLE |
| | THI | $15.69 \pm 3.44$ | TLE | TLE | $10.73 \pm 1.48$ | $19.10 \pm 0.96$ | TLE |

(b) Results for regression datasets. $R^2$ is reported for all datasets.

| | | hm-prices | avazu-ctr | twitch-views | artnet-views |
|---|---|---|---|---|---|
| best const. pred. | RH | $0.00 \pm 0.00$ | $0.00 \pm 0.00$ | $0.00 \pm 0.00$ | $0.00 \pm 0.00$ |
| | TH | $-2.85 \pm 0.00$ | $0.00 \pm 0.00$ | $-22.31 \pm 0.00$ | $-9.32 \pm 0.00$ |
| | THI | $-2.85 \pm 0.00$ | $0.00 \pm 0.00$ | $-22.31 \pm 0.00$ | $-9.32 \pm 0.00$ |
| LightGBM | RH | $73.99 \pm 0.11$ | $29.60 \pm 0.02$ | $13.37 \pm 0.00$ | $36.38 \pm 0.08$ |
| | TH | $63.56 \pm 0.39$ | $26.23 \pm 0.04$ | $-9.60 \pm 0.03$ | $42.87 \pm 0.08$ |
| | THI | $63.56 \pm 0.39$ | $26.23 \pm 0.04$ | $-9.60 \pm 0.03$ | $42.87 \pm 0.08$ |
| LightGBM-NFA | RH | $79.78 \pm 0.09$ | $35.35 \pm 0.04$ | $62.14 \pm 0.01$ | $60.30 \pm 0.07$ |
| | TH | $71.36 \pm 0.41$ | $38.69 \pm 0.03$ | $43.60 \pm 0.03$ | $52.42 \pm 0.06$ |
| | THI | $68.88 \pm 0.11$ | $33.04 \pm 0.17$ | $24.81 \pm 0.14$ | $51.95 \pm 0.05$ |
| GCN | RH | $79.76 \pm 0.76$ | $34.96 \pm 0.11$ | $77.12 \pm 0.11$ | $61.02 \pm 0.13$ |
| | TH | $65.20 \pm 0.84$ | $37.49 \pm 0.26$ | $68.17 \pm 0.24$ | $54.44 \pm 0.43$ |
| | THI | $64.31 \pm 0.82$ | $34.78 \pm 0.48$ | $63.58 \pm 0.54$ | $53.73 \pm 0.47$ |
| GraphSAGE | RH | $79.89 \pm 0.46$ | $35.20 \pm 0.20$ | $72.02 \pm 0.16$ | $56.65 \pm 0.50$ |
| | TH | $67.93 \pm 1.24$ | $38.38 \pm 0.39$ | $61.46 \pm 0.68$ | $51.87 \pm 0.48$ |
| | THI | $65.80 \pm 0.56$ | $36.79 \pm 0.55$ | $56.60 \pm 0.71$ | $53.37 \pm 0.30$ |
| GAT | RH | $81.68 \pm 0.41$ | $35.74 \pm 0.19$ | $76.06 \pm 0.30$ | $59.01 \pm 0.52$ |
| | TH | $70.83 \pm 0.99$ | $39.21 \pm 0.17$ | $66.32 \pm 0.59$ | $52.30 \pm 0.34$ |
| | THI | $69.74 \pm 1.50$ | $37.18 \pm 1.02$ | $61.41 \pm 1.30$ | $52.44 \pm 0.59$ |
| GT | RH | $80.90 \pm 0.42$ | $34.38 \pm 0.31$ | $75.57 \pm 0.15$ | $58.97 \pm 0.25$ |
| | TH | $69.70 \pm 0.84$ | $38.27 \pm 0.27$ | $65.83 \pm 0.24$ | $51.67 \pm 0.54$ |
| | THI | $67.33 \pm 2.05$ | $36.49 \pm 0.83$ | $60.67 \pm 1.02$ | $52.26 \pm 0.48$ |

Further, we can see that while there is no single best model among GNNs, attention-based GNNs (GAT and GT) outperform classic GNNs (GCN and GraphSAGE) on most datasets, showing the importance of being able to assign weights to messages from graph neighbors depending on their content in realistic industrial datasets.

As for GFMs, we find that all the considered models produce very weak results on all our datasets. This shows that current GFMs are still far from being able to compete with classic GNNs on realistic datasets with rich node attributes.

### 5.3 High label rate, random and temporal data splits, transductive and inductive settings

A summary of the results for high label rate random (RH) and temporal (TH) data splits, as well as for the inductive setting (THI) is provided in Table 3. Full results are provided in Appendix C.

First, the observations from the previous subsection about the usefulness of graph structure, benefits of attention-based GNNs, and weak performance of GFMs also apply to high label rate settings. Next, we observe that temporal data splits are significantly more challenging for all models than random ones (with the exception of the `avazu-ctr` dataset). This is important as in real-world applications temporal distributional shifts are common, and not considering them can provide overly optimistic performance estimates. Further, the considered models perform significantly worse in the inductive setting than in the transductive one. These observations highlight the importance of developing GML methods that are more resilient to temporal distributional shifts and dynamic changes in the graph structure for industrial applications. In the absence of such methods, frequent retraining of GNNs on new data is recommended to achieve the best results. Note that GFMs that only utilize in-context learning to adapt to new graphs do not suffer from the transductive/inductive mismatch and thus represent a promising direction, but their performance is currently very weak compared with GNNs on all datasets.

Overall, our main findings are as follows:

- GNNs can provide substantial benefits in industrial applications where graph information is available, with attention-based GNNs showing particularly strong results. However, GBDT models popular in industrial applications can serve as strong baselines when provided with graph information as additional input features, especially in regression tasks.

- GML methods can be very strongly affected by temporal distributional shifts and dynamically evolving graphs, so the design of GML methods capable of better handling such settings is an important research direction.

- The current general-purpose graph foundation models achieve very weak results on our datasets. Thus, the problem of creating truly general GFMs is far from being solved. We hope our benchmark can serve as a reliable testbed for future research in this direction. The first such successful uses have already appeared: after we made GraphLand publicly available, Eremeev et al. (2025a,b) introduced the first GFMs that achieve strong results on our datasets.

## 6   Conclusion

We introduce GraphLand — a set of diverse graph datasets representing realistic industrial applications of GML. Besides serving as an extended testbed for GNNs and GFMs, GraphLand allows investigating such previously underexplored research questions in the GML literature as how temporal distributional shifts and the inductive prediction setting influence the performance of GML methods. We hope GraphLand will encourage the evaluation of GML methods under more realistic and diverse settings, the development of GML methods that are more resilient to temporal distributional shifts and dynamically changing graph structure, and the development of better performing GFMs that can handle different node feature sets that go beyond textual descriptions.

### Acknowledgments

We thank Islam Umarov for providing the data for `city-reviews` dataset; Daniil Fedulov and Daniil Bondarenko for collecting the data for `artnet-views` and `artnet-exp` datasets; Dmitry Bikulov and Alexander Ulyanov for sharing the data for `web-traffic`, `web-fraud`, and `web-topics` datasets; Alexandr Ruchkin and Aleksei Istomin for collecting the data for `city-roads-M` and `city-roads-L` datasets; Fedor Velikonivtsev for helping with data processing for `city-roads-M` and `city-roads-L` datasets. We also thank Dmitry Eremeev for useful discussions.

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

# A GraphLand benchmark details

## A.1 Dataset description

In this section, we provide a more detailed description of the GraphLand datasets. Note that none of the proposed datasets contain any personal information. For the datasets based on data newly collected for our benchmark, the public release of the data was approved by a legal team. GraphLand datasets are available at Zenodo and Kaggle. Our source code and instructions on how to reproduce our experimental results are available in our GitHub repository.

**web-fraud**, **web-topics**, and **web-traffic** These three datasets are web-graphs — they represent a segment of the Internet. The nodes are websites, and a directed edge connects two nodes if at least one user followed a link from one website to the other in a selected period of time. We prepared three datasets with the same graph but different tasks: in web-fraud, the task is to predict which websites are fraudulent (strongly imbalanced binary classification); in web-topics, the task is to predict the topic that a website belongs to (multiclass classification); and in web-traffic, the task is to predict how many users visited a website in a specific period of time (regression). With almost 3 million nodes, this is one of the largest publicly available attributed graphs that is not a citation network. Nodes in this graph have more than two hundred features, examples of which include the number of videos on the website (numerical feature), the website's zone, and whether the website is on a free hosting (categorical features). Data for these datasets was obtained from the Yandex search engine.

**artnet-views** and **artnet-exp** These two datasets represent a social network of art creators. The nodes are users, and an undirected edge connects two nodes if the users are friends. We prepared two datasets with the same graph but different tasks: in artnet-views, the task is to predict how many views a user receives in a specific period of time (regression); and in artnet-exp, the task is to predict which users create explicit art content (binary classification). Examples of node features include user interests (categorical features). Data for these datasets was obtained from the generative AI art creation platform Shedevrum.

**city-roads-M** and **city-roads-L** These datasets are obtained from the logs of a navigation service and represent the road networks of two major cities, with the second one being several times larger than the first. The nodes are segments of roads, and a directed edge connects two nodes if the segments are incident to each other and moving from one segment to the other is permitted by traffic rules. The task is to predict the average travel speed on a road segment at a specific timestamp (regression). The features include various information about the road segment such as binary indicators of whether there is a bike dismount sign, whether the road segment ends with a crosswalk or a toll post, whether the road segment is in poor condition, whether it is restricted for trucks, and whether it has a mass transit lane (categorical features). Numerical features include the length of the road segment and the geographic coordinates of the road endpoints. Data for these datasets was obtained from the Yandex Maps service.

**city-reviews** This dataset is obtained from the logs of a review service in which users can leave reviews and ratings for places and organizations in two major cities. The nodes are users, and an undirected edge connects two nodes if the users often leave reviews for the same organizations. The task is fraud detection — to predict which users leave fraudulent reviews (binary classification). The node features are based on user interactions with the service and their examples include the share of negative reviews among all reviews left by a user (numerical feature) and the browser that is used to access the service by the user (categorical feature). Data for this dataset was obtained from the Yandex Maps service.

**avazu-ctr** This dataset is based on open data that has been introduced at the Kaggle competition organized by Avazu (Wang & Cukierski, 2014). The data contains information about interactions between devices used to access the Internet, websites, and advertisements. In our graph, the nodes are devices, and an edge connects two nodes if the devices often visit the same websites. The graph is undirected. A smaller version of a similar dataset has been used by Ivanov & Prokhorenkova (2021); however, it contained only a small subset of devices, while for our dataset we collected data for all the available devices, making our graph more than 50 times larger. The task is to predict the advertisement click-through rate (CTR) observed on devices (regression). Nodes in this graph have more than two hundred numerical features; however, most of them were anonymized in the original data source.

**hm-categories** and **hm-prices** These datasets are based on open data that has been introduced at the Kaggle competition organized by H&M (García Ling et al., 2022). The graph represents a co-purchasing network. The nodes are products, and an edge connects two nodes if the products are often bought by the same customers. The graph is undirected. We prepared two datasets with the same graph but different tasks: in `hm-categories`, the task is to predict the product category (multiclass classification), and in `hm-prices`, the task is to predict the product price (regression). The node features in this dataset include product metadata such as product color (categorical feature), as well as information obtained from product purchasing statistics such as what proportion of product purchases occurs on different weekdays (numerical features).

**pokec-regions** This dataset is based on the data from Takac & Zabovsky (2012). It represents the online social network Pokec. The nodes are users, and a directed edge connects two nodes if one user has marked the other one as a friend. While this graph is quite popular in network analysis as an example of a classic social network, it is relatively rarely used for machine learning, with the exception of Lim et al. (2021) who use the same graph as we do but with different task, node features, and data split. In our dataset, the task is to predict which region a user is from (extreme multiclass classification with 183 classes). The node features in our dataset are based on user profile information, examples of them include the profile completion proportion (numerical feature) and binary indicators of whether different profile fields are filled (categorical features).

**twitch-views** This dataset is based on the data from Rozemberczki & Sarkar (2021). It represents the live-streaming network Twitch. The nodes are users, and an undirected edge connects two nodes if both users follow each other. The task is to predict how many views a user receives in a specific period of time (regression). The node features are based on user profile information and examples of them include user language and affiliate status (categorical features).

**tolokers-2** This is a new version of the dataset `tolokers` from Platonov et al. (2023b); Likhobaba et al. (2023) with a significantly extended set of node features. It is based on the data from the Toloka crowdsourcing platform and the graph represents a network of tolokers (workers). The nodes are tolokers, and an edge connects two nodes if these tolokers have worked on the same task. The graph is undirected. The task is fraud detection — to predict which tolokers have been banned in one of the projects (binary classification). The new node features include various performance statistics of workers, such as the number of approved assignments and the number of skipped assignments (numerical features), as well as workers' profile information, such as their education level (categorical feature).

Some graphs in our benchmark are undirected, and some are directed (see individual dataset descriptions above). All our undirected graphs consist of a single connected component, and all our directed graphs consist of a single weakly connected component.

For all datasets, we provide random stratified RL and RH data splits. Further, we provide temporal TH data split (with the possibility of using the inductive learning setting THI) for all datasets with the exception of `city-roads-M` and `city-roads-L` datasets (since well-established road network graphs typically do not evolve over time significantly), as well as `city-reviews` and `web-traffic` datasets (since for them some of the necessary temporal information was not available).

### A.2    Dataset properties

A key characteristic of our benchmark is its diversity. As described above, our graphs come from different domains and have different prediction tasks. Their edges are also constructed in different ways (based on user interactions, activity similarity, physical connections, etc.). However, the proposed datasets also differ in many other ways. Some properties of our graphs are presented in Table 1 (see below for the details on how the provided characteristics are defined). First, note that the sizes of our datasets range from 11K to 3M nodes. The smaller graphs can be suitable for compute-intensive models, while the larger graphs can provide a moderate scaling challenge. The average and median degrees of our graphs also vary significantly and our benchmark has both sparse and relatively dense graphs, including graphs with the average degree in the order of hundreds which is larger than the average degrees of most datasets used in current GML research (such graphs may highlight the importance of attention-based GNNs with their soft edge selection mechanisms). The average distance between two nodes in our graphs varies from $2.45$ for `hm-categories` and `hm-prices` to $194$ for `city-roads-L`; and graph diameter (maximum distance) varies from $8$ for

`twitch-views` to $553$ for `city-roads-L`. Further, we report the values of clustering coefficients, which show how typical closed node triplets are for the graph. In the literature, there are two definitions of clustering coefficients (Boccaletti et al., 2014): the global clustering coefficient and the average local clustering coefficient. We have both graphs where the clustering coefficients are high and graphs where they are almost zero, as well as graphs where global and local clustering coefficients significantly disagree (which is possible for graphs with imbalanced degree distributions). The degree assortativity coefficient is defined as the Pearson correlation coefficient of degrees among pairs of linked nodes. For most of our graphs, the degree assortativity is either negative or close to zero, which means that nodes do not tend to connect to other nodes with similar degrees, while `city-roads-M` and `city-roads-L` datasets are the exceptions — for them the degree assortativity is positive and large.

Further, let us discuss the graph-label relationships in our datasets. To measure the similarity of labels of connected nodes for regression datasets, we use target assortativity — the Pearson correlation coefficient of target values between pairs of connected nodes. For instance, for the `city-roads-M` and `city-roads-L` datasets, the target assortativity is positive and quite large, which shows that nodes tend to connect to other nodes with similar target values (which is expected for the task of speed prediction in road networks), while for the `twitch-views` and `web-traffic` datasets, the target assortativity is negative. For classification datasets, the similarity of neighbors' labels is usually called *homophily*: in homophilous datasets, nodes tend to connect to nodes of the same class. How to properly measure homophily has recently attracted some research. It has been noted by Lim et al. (2021) and Platonov et al. (2023a) that homophily measures typically used in the literature — such as the proportion of edges connecting nodes of the same class — are not appropriate for comparing homophily levels between graphs with different numbers of classes and their size balance. Platonov et al. (2023a) proposed a set of properties that a homophily measure appropriate for use in such comparisons should satisfy and Mironov & Prokhorenkova (2024) constructed the first known homophily measure that satisfies all these properties — *unbiased homophily*. Thus, in our work, we use unbiased homophily to measure the homophily levels of our datasets. Unbiased homophily (with $\alpha = 0$, see Mironov & Prokhorenkova (2024) for more details) takes values in $[-1, \ 1]$ with $1$ indicating perfect homophily, $-1$ indicating perfect heterophily, and $0$ indicating no preference between homophilous and heterophilous edges (such graphs are typically referred to as heterophilous in the literature, although a more appropriate term would be non-homophilous). Note that the values of unbiased homophily should not be compared to values of other homophily measures used in the literature; the unbiased homophily levels for some popular graph node classification datasets are provided by Mironov & Prokhorenkova (2024). Unbiased homophily indicates that among our datasets `pokec-regions` and `city-reviews` are homophilous, while the remaining datasets are non-homophilous. Thus, our benchmark significantly expands the set of available non-homophilous graph datasets.

Finally, our datasets have diverse sets of node features consisting of numerical and categorical features with different meanings and distributions. All our datasets except `twitch-views` and `tolokers-2` have at least several dozen node features, while some have several hundred node features.

Overall, our datasets are diverse in domain, scale, structural properties, graph-label relations, and node attributes. Coming from real-world GML applications, they may serve as a valuable tool for the research and development of GML methods for the industry.

**Computing dataset characteristics** Further, we describe the characteristics that are used in Table 1. Note that, while some graphs in our benchmark are directed, we transformed all the graphs to be undirected before computing all the considered graph characteristics, since some of the characteristics are not defined for directed graphs.

*Average degree* and *median degree* are the average and median numbers of neighbors a node has, respectively. Since all our graphs are connected (when treated as undirected graphs), for any two nodes there is a path between them. *Average distance* is the average length of the shortest paths between all pairs of nodes, while *diameter* is the maximum length of the shortest paths between all pairs of nodes. For our largest graphs — the ones used for the `pokec-regions`, `web-traffic`, `web-fraud`, and `web-topics` datasets — we approximate the average distance with the average over distances for 100K randomly sampled node pairs. *Global clustering* coefficient is computed as three times the number of triangles divided by the number of pairs of adjacent edges (i.e., it is the fraction of closed triplets of nodes among all connected triplets). *Average local clustering*

coefficient first computes the local clustering of each node, which is the fraction of connected pairs of its neighbors, and then averages the obtained values among all nodes. *Degree assortativity* is the Pearson correlation coefficient between the degrees of connected nodes. Further, *target assortativity* for regression datasets is the Pearson correlation coefficient between target values of connected nodes. For computing *unbiased homophily*, we follow Mironov & Prokhorenkova (2024) and use the simplest version of this measure with the $\alpha$ parameter set to 0.

# B   Experimental setup

## B.1   Models

**Graph-agnostic models**   As a simple baseline, we use ResMLP — an MLP with skip-connections (He et al., 2016) and layer normalization (Ba et al., 2016). This model does not have any information about the graph structure and operates on nodes as independent samples — we call such models *graph-agnostic*. It has been shown that such MLP-like models with skip-connections can serve as very strong baselines for industrial data with mixed numerical and categorical features (Gorishniy et al., 2021). Further, we consider the three most popular implementations of GBDT models that are widely used in industrial applications: XGBoost (Chen & Guestrin, 2016), LightGBM (Ke et al., 2017), and CatBoost (Prokhorenkova et al., 2018). GBDT models are often considered to be better than neural networks at dealing with numerical features (Gorishniy et al., 2021, 2022) and regression tasks (Zhang et al., 2023).

The models discussed above are graph-agnostic. To see if they can be improved by being granted access to some of the graph information, we augment them with *neighborhood feature aggregation* (NFA) — a simple feature augmentation technique that extends features of each graph node with aggregated information about features of its neighbors (see Appendix B.2 for a detailed discussion of NFA). Specifically, we add NFA to ResMLP and LightGBM, since they are our fastest graph-agnostic models. We denote these model versions with the -NFA suffix. We expect that if the graph structure is beneficial for the considered task, then NFA-augmented models will significantly outperform their counterparts without NFA. This indeed happens in our experiments, confirming the usefulness of the graph structure provided in our datasets. However, note that NFA provides only limited access to graph information, specifically the aggregated features of 1-hop neighbors. Models that can use much more graph information are graph neural networks, which we discuss in the next paragraph.

**Graph neural networks**   We consider several representative GNN architectures. First, we use GCN (Kipf & Welling, 2017) and GraphSAGE (Hamilton et al., 2017) as simple classical GNN models. For GraphSAGE, we use the version with the mean aggregation function, and we do not use the neighbor sampling technique proposed in the original paper, instead training the model on the full graph, like all other GNNs in our experiments. Further, we use two GNNs with attention-based neighborhood aggregation functions: GAT (Veličković et al., 2018) and Graph Transformer (GT) (Shi et al., 2021). Note that GT is a *local* graph transformer, i.e., each node only attends to its neighbors in the graph (in contrast to *global* graph transformers, in which each node attends to all other nodes in the graph, and which are thus not instances of the standard message-passing neural networks (MPNNs) framework of Gilmer et al. (2017)). Following Platonov et al. (2023b), we equip all the considered GNNs with skip-connections and layer normalization, which we found important for their strong performance on our datasets. We also add a two-layer MLP with the GELU activation function (Hendrycks & Gimpel, 2016) after every neighborhood aggregation block in GNNs. Our graph models are implemented in the same codebase as our ResMLP — we simply swap each residual block of ResMLP with a residual neighborhood aggregation block of the selected GNN architecture. Therefore, comparing the performance of ResMLP and GNNs is one more way to see if graph information is helpful for the task. Indeed, in our experiments, GNNs significantly outperform ResMLP on all our datasets, once again confirming the usefulness of the provided graph structure for the considered tasks.

**Graph foundation models**   Most current GFMs do not support node property prediction tasks in graphs with arbitrary node features. Among those that do, we were able to find only two models with open pretrained checkpoints: OpenGraph (Xia et al., 2024) and AnyGraph (Xia & Huang, 2024). Both OpenGraph and AnyGraph use the Transformer architecture and are pretrained with a link prediction objective on a mixture of different graph datasets. These methods differ in what data they can operate on. Specifically, OpenGraph only uses relational information and constructs

node representations based on SVD factors of the adjacency matrix, while AnyGraph also uses the available node feature information and combines SVD factors for both the feature matrix and the adjacency matrix. Both these models were designed to be adapted to new node classification datasets without fine-tuning, using an in-context learning (ICL) setting. Specifically, they can perform link prediction in arbitrary graphs, and they cast any node classification task as a link prediction task where links to virtual nodes representing classes are predicted for unlabeled nodes.

Further, we reproduced the pretraining for two more GFMs — TS-GNN (Finkelshtein et al., 2025) and GCOPE (Zhao et al., 2024) (the weights for these models are not publicly available, but the training code is). GCOPE also applies a projection to node features (e.g., based on SVD or an attention mechanism), and also uses additional virtual nodes as graph coordinators to simultaneously process different graph datasets at the pretraining stage. The authors use GraphCL (You et al., 2020) or SimGRACE (Xia et al., 2022) as the pretraining objective. In contrast to the ICL approaches, GCOPE uses fine-tuning for adaptation to new node classification datasets.

The TS-GNN model takes a very different approach. Its authors observe that a GFM is expected to be feature-permutation-invariant, label-permutation-equivariant, and node-permutation-equivariant, design a linear transformation that satisfies these properties, and use it as the main GFM building block. Note that TS-GNN requires solving several least squares problems to fit its weights before making predictions, which is a form of training, but, following the authors of this model, we still mark it as ICL due to these least squares problems being computationally light compared to typical fine-tuning. The original paper proposes two versions of TS-GNN: TS-Mean and TS-GAT; we use TS-Mean in all our experiments.

Note that neither of the considered GFMs supports node regression (for OpenGraph and AnyGraph this is an inherent limitation of the model design, while for TS-GNN and GCOPE this is only a limitation of the official model implementations, since in theory these two models can perform node regression). Our experiments also show that these models cannot scale to large datasets. These two issues prevent us from successfully evaluating the considered GFMs on a significant number of our datasets.

## B.2 Neighborhood feature aggregation

Below we describe our NFA technique. This technique augments node features with the information about features of the node's neighbors in the graph. As we show in our experiments, this technique significantly improves the performance of originally graph-agnostic models on our datasets. We consider the set of 1-hop neighbors of each node and compute various statistics over the node features in this set. In particular, for numerical features, we compute their mean, maximum, and minimum values in the neighborhood excluding any `NaN`s. If all neighbor values of a particular feature are `NaN`s, we fill the corresponding statistics with `NaN`s as well. For categorical features, we first transform them into a set of binary features using one-hot encoding. Then, for each binary feature, we compute the mean value in the neighborhood, i.e., the ratio of 1s for the binary indicator. Then, we concatenate all the produced additional features with the original node features. More formally, consider some specific feature $x \in X$ from the set of features $X$, an arbitrary node $v \in V$ in the graph $G(V, E)$, and its 1-hop neighbors $\mathcal{N}_G(v)$ that do not contain `NaN` in feature $x$. Then, we can collect the set $S$ of non-`NaN` values $x_u$ from its neighbors $u \in \mathcal{N}_G(v)$ and apply some permutation-invariant aggregation function $f$ to them in order to obtain a single value $h$:

$$S = \big\{x_u : u \in \mathcal{N}_G(v) \wedge x_u \text{ is not } \texttt{NaN}\big\}, \ f(S) = h.$$

Note that $f(\varnothing) = \texttt{NaN}$. This value $h$ is then used as an additional feature for the considered node $v$. This procedure is done for each node $v \in V$ and each feature $x \in X$. In particular, for numerical features, we apply three aggregation functions separately: `mean`, `max`, `min`, thus producing three new features. For categorical features, we first apply one-hot encoding to them, and then apply the `mean` aggregation function to each of the resulting binary features, thus producing as many additional features as there were possible values of the original categorical feature. We concatenate this NFA vector to the vector of the original features of the node $v$.

In Figure 1, we provide a simple example of applying NFA. Here, we consider a central node with four neighbors, which have one numerical feature (blue) and one categorical feature (green). To construct NFA for the central node, we compute `mean`, `max`, `min` values for the numerical feature and `mean` value for the one-hot-encoded categorical feature across all its neighboring nodes.

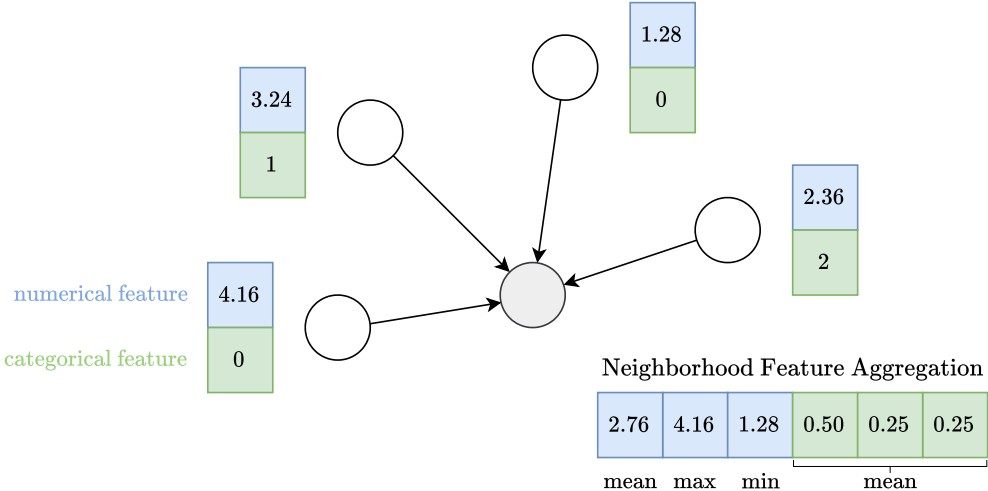

Figure 1: An example of applying neighborhood feature aggregation (NFA).

## B.3 Experimental setup and hyperparameter selection details

Some of the graphs in our benchmark are directed. For our experiments, we converted directed graphs to undirected ones (by replacing each directed edge with an undirected one and then removing duplicated edges). We leave the investigation of different ways to consider edge directions to further research.

We train all models 10 times with different random seeds to compute the mean and standard deviation of model performance, except for our largest datasets `pokec-regions`, `web-traffic`, `web-fraud`, `web-topics`, for which we train all models 5 times. We train all our GNNs in a full-batch setting, i.e., we do not use any subgraph sampling techniques and train the models on the full graph. Our ResMLP baseline is implemented in the same codebase as our GNNs and thus is also trained in the full-batch setting. All the experiments were run on an NVIDIA Tesla A100 80GB GPU, except for GBDTs, which were trained on AMD EPYC CPUs.

Hyperparameter choice is extremely important for the performance of both GNNs and GBDT models. Thus, we conducted an extensive hyperparameter search on the validation set for all models. For the considered GBDT models, we ran 100 iterations of Bayesian optimization using Optuna (Akiba et al., 2019). The specific hyperparameter distributions used for these models are provided in Table 4. Since GNNs are sensitive to different hyperparameters than GBDT models, we used a different hyperparameter search strategy for them. Specifically, we found that the learning rate and dropout probability (Srivastava et al., 2014) are the most important hyperparameters for our GNN implementations on our datasets. Thus, we ran grid search selecting the learning rate from $\{3 \times 10^{-5}, 1 \times 10^{-4}, 3 \times 10^{-4}, 1 \times 10^{-3}, 3 \times 10^{-3}\}$ and dropout probability from $\{0, 0.1, 0.2\}$ (note that the highest learning rate of $3 \times 10^{-3}$ often resulted in `NaN` issues; however, we still included it in our hyperparameter search, as in our preliminary experiments we found it to be beneficial for some of our dataset/model combinations). In our preliminary experiments we found that the performance of our GNNs is quite stable for a wide variety of reasonable architecture hyperparameter values (we found the use of skip-connections and layer normalization to be important for this stability). Hence, for our final experiments, we kept these hyperparameters fixed. We set these values as follows: the number of graph neighborhood aggregation blocks to 3 and the hidden dimension to 512. The only exceptions to this hidden dimension size were made for our largest datasets: to avoid GPU out-of-memory issues, we decreased the hidden dimension to 400 for `pokec-regions` and to 200 for `web-traffic`, `web-fraud`, and `web-topics`. For GNNs with attention-based graph neighborhood aggregation (GAT and GT), the number of attention heads was set to 4. We used the Adam optimizer (Kingma & Ba, 2015) in all our GNN experiments. We trained each model for 1000 steps and then selected the best step based on the performance on the validation set.

When applying deep learning models to data with numerical features, the preprocessing of these features is critically important. In our experiments, we considered two possible numerical feature

Table 4: The Optuna hyperparameter search distributions for GBDT models.

| XGBoost | | LightGBM | | CatBoost | |
|---|---|---|---|---|---|
| Parameter | Distribution | Parameter | Distribution | Parameter | Distribution |
| colsample_bytree | Uniform[0.5, 1.0] | feature_fraction | Uniform[0.5, 1.0] | bagging_temperature | Uniform[0.0, 1.0] |
| gamma | {0.0, LogUniform[0.001, 100.0]} | lambda_l2 | {0.0, LogUniform[0.1, 10.0]} | depth | UniformInt[3, 14] |
| lambda | {0.0, LogUniform[0.1, 10.0]} | learning_rate | LogUniform[0.001, 1.0] | l2_leaf_reg | Uniform[0.1, 10.0] |
| learning_rate | LogUniform[0.001, 1.0] | num_leaves | UniformInt[4, 768] | leaf_estimation_iterations | Uniform[1, 10] |
| max_depth | UniformInt[3, 14] | min_sum_hessian_in_leaf | LogUniform[0.0001, 100.0] | learning_rate | LogUniform[0.001, 1.0] |
| min_child_weight | LogUniform[0.0001, 100.0] | bagging_fraction | Uniform[0.5, 1.0] | | |
| subsample | Uniform[0.5, 1.0] | | | | |

transformation techniques: standard scaling and quantile transformation to standard normal distribution. We included them in the hyperparameter search for ResMLP and GNNs. In contrast, GBDT models do not need specialized preprocessing for numerical features and are not affected by their monotonic transformations. For categorical features, we used one-hot encoding for all models except for LightGBM and CatBoost, which support the use of categorical features directly and have their specialized strategies for working with them (XGBoost also offers such a feature, but it is currently marked as experimental, and we were not able to make it work). For regression datasets, neural models might perform better if the target variable is transformed. Therefore, in our experiments on regression datasets with ResMLP and GNNs, we considered the options of using the original targets or preprocessing targets with standard scaling, including these two options in the hyperparameter search.

Our GNNs are implemented using PyTorch (Paszke et al., 2019) and DGL (Wang et al., 2019).

## C   Complete experimental results

In the main text, we report our complete experimental results for the RL setting, but, due to space limitations, for the RH, TH, THI settings, we leave out results for some of the baselines (ResMLP, XGBoost, CatBoost, ResMLP-NFA) and for the datasets that do not have temporal data split (city-roads-M, city-roads-L, city-reviews, web-traffic). We provide complete experimental results for the RH setting in Table 5, for the TH setting in Table 6, and for the THI setting in Table 7. In all tables with results, for each dataset, we highlight with color the best result as well as those results for which the mean differs from the best one by no more than the sum of the two results' standard deviations.

## D   Limitations and broader impact

The aim of our benchmark is to introduce a diverse set of graph datasets for node property prediction that covers a wide range of domains and graph structural properties, including those not encountered in commonly used datasets for GML model evaluation. However, data that can be naturally represented as graphs is so widespread across different domains that no benchmark can cover them all. Thus, our collection of 14 datasets still only covers a small part of the wide range of situations where modeling data as a graph can be useful. Nevertheless, we hope that it will encourage the GML research community to use more diverse sets of datasets and focus on practically relevant applications where graph-structured data appears.

Our benchmark includes datasets with realistic tasks such as fraud detection and user engagement prediction. Poorly performing machine learning models used for these tasks in real-world services can negatively affect the users of these services. For example, type I errors of fraud detection systems, i.e., wrongly predicting that an innocent person is fraudulent, have an undesirable negative impact. Thus, particular care should be taken to minimize the probability of such errors. We believe that the release of high-quality and properly anonymized datasets for these tasks such as those in our benchmark will encourage the community to develop better models, since the community will be able to use these datasets as a realistic and reliable testbed to investigate which methods lead to reductions in undesirable model errors.

Table 5: Experimental results under the `RH` (random high) data split in the transductive setting. The best result and those statistically indistinguishable from it are highlighted in red. `TLE` stands for time limit exceeded (24 hours); `MLE` stands for memory limit exceeded (80 GB VRAM); `RTE` stands for runtime error in the official code of GFMs.

(a) Results for classification datasets. Accuracy is reported for multiclass classification datasets and Average Precision is reported for binary classification datasets.

| | multiclass classification | | | binary classification | | | |
|---|---|---|---|---|---|---|---|
| | hm-categories | pokec-regions | web-topics | tolokers-2 | city-reviews | artnet-exp | web-fraud |
| best const. pred. | $19.46 \pm 0.00$ | $3.77 \pm 0.00$ | $28.36 \pm 0.00$ | $21.82 \pm 0.00$ | $12.09 \pm 0.00$ | $10.00 \pm 0.00$ | $0.66 \pm 0.00$ |
| ResMLP | $43.12 \pm 0.25$ | $5.09 \pm 0.01$ | $44.55 \pm 0.08$ | $45.96 \pm 0.46$ | $75.21 \pm 0.08$ | $43.55 \pm 0.23$ | $13.52 \pm 0.21$ |
| XGBoost | $46.68 \pm 0.13$ | $5.11 \pm 0.01$ | TLE | $49.31 \pm 1.05$ | $77.77 \pm 0.13$ | $45.34 \pm 0.40$ | $16.06 \pm 0.21$ |
| LightGBM | $47.28 \pm 0.14$ | $5.10 \pm 0.01$ | TLE | $49.92 \pm 0.19$ | $77.88 \pm 0.13$ | $44.98 \pm 0.80$ | TLE |
| CatBoost | $46.98 \pm 0.24$ | TLE | TLE | $50.52 \pm 0.19$ | $78.00 \pm 0.05$ | $45.50 \pm 0.15$ | TLE |
| ResMLP-NFA | $61.34 \pm 0.34$ | $12.24 \pm 0.12$ | MLE | $56.77 \pm 1.15$ | $79.80 \pm 0.07$ | $44.52 \pm 0.44$ | MLE |
| LightGBM-NFA | $67.09 \pm 0.15$ | $12.15 \pm 0.02$ | TLE | $62.19 \pm 0.35$ | $81.63 \pm 0.06$ | $49.26 \pm 0.18$ | TLE |
| GCN | $73.38 \pm 0.42$ | $35.08 \pm 0.62$ | $48.88 \pm 0.09$ | $60.49 \pm 0.86$ | $81.05 \pm 0.10$ | $49.80 \pm 0.35$ | $15.58 \pm 0.20$ |
| GraphSAGE | $73.34 \pm 0.68$ | $40.76 \pm 0.21$ | $50.05 \pm 0.03$ | $58.42 \pm 0.92$ | $80.75 \pm 0.06$ | $48.49 \pm 0.37$ | $20.47 \pm 0.17$ |
| GAT | $79.19 \pm 0.21$ | $46.72 \pm 0.69$ | $50.54 \pm 0.04$ | $63.76 \pm 1.30$ | $81.10 \pm 0.11$ | $50.62 \pm 0.35$ | $20.43 \pm 0.21$ |
| GT | $79.28 \pm 0.31$ | $50.06 \pm 0.53$ | $50.58 \pm 0.04$ | $60.32 \pm 1.21$ | $80.50 \pm 0.14$ | $49.32 \pm 1.00$ | $19.73 \pm 0.34$ |
| OpenGraph (ICL) | $11.69 \pm 0.84$ | $2.56 \pm 0.42$ | RTE | $44.62 \pm 1.35$ | $62.96 \pm 0.84$ | $23.72 \pm 1.86$ | RTE |
| AnyGraph (ICL) | $15.65 \pm 2.82$ | $27.67 \pm 2.48$ | $6.30 \pm 2.82$ | $30.21 \pm 3.32$ | $65.04 \pm 1.41$ | $15.80 \pm 1.43$ | $0.67 \pm 0.02$ |
| TS-GNN (ICL) | $15.48 \pm 1.93$ | MLE | MLE | $31.83 \pm 2.55$ | $11.84 \pm 1.98$ | $13.38 \pm 2.59$ | MLE |
| GCOPE (FT) | $19.99 \pm 0.12$ | TLE | TLE | $31.79 \pm 1.95$ | $69.74 \pm 0.36$ | $23.86 \pm 2.33$ | TLE |

(b) Results for regression datasets. $R^2$ is reported for all datasets.

| | hm-prices | avazu-ctr | city-roads-M | city-roads-L | twitch-views | artnet-views | web-traffic |
|---|---|---|---|---|---|---|---|
| best const. pred. | $0.00 \pm 0.00$ | $0.00 \pm 0.00$ | $0.00 \pm 0.00$ | $0.00 \pm 0.00$ | $0.00 \pm 0.00$ | $0.00 \pm 0.00$ | $0.00 \pm 0.00$ |
| ResMLP | $70.11 \pm 0.48$ | $28.03 \pm 0.22$ | $62.43 \pm 0.32$ | $53.09 \pm 0.17$ | $13.36 \pm 0.01$ | $36.10 \pm 0.17$ | $73.88 \pm 0.05$ |
| XGBoost | $74.49 \pm 0.14$ | $29.70 \pm 0.05$ | $70.93 \pm 0.05$ | $64.62 \pm 0.07$ | $13.34 \pm 0.02$ | $36.83 \pm 0.06$ | TLE |
| LightGBM | $73.99 \pm 0.11$ | $29.60 \pm 0.02$ | $70.01 \pm 0.19$ | $63.66 \pm 0.09$ | $13.37 \pm 0.00$ | $36.38 \pm 0.08$ | TLE |
| CatBoost | $74.92 \pm 0.14$ | $29.68 \pm 0.10$ | $69.32 \pm 0.17$ | $61.24 \pm 0.11$ | $13.25 \pm 0.03$ | $37.47 \pm 0.05$ | TLE |
| ResMLP-NFA | $74.99 \pm 0.50$ | $34.93 \pm 0.21$ | $65.87 \pm 0.31$ | $57.96 \pm 0.15$ | $56.97 \pm 0.28$ | $54.95 \pm 0.55$ | MLE |
| LightGBM-NFA | $79.78 \pm 0.09$ | $35.35 \pm 0.04$ | $72.09 \pm 0.07$ | $66.24 \pm 0.08$ | $62.14 \pm 0.01$ | $60.30 \pm 0.07$ | TLE |
| GCN | $79.76 \pm 0.76$ | $34.96 \pm 0.11$ | $69.95 \pm 0.11$ | $64.65 \pm 0.27$ | $77.12 \pm 0.11$ | $61.02 \pm 0.13$ | $83.49 \pm 0.14$ |
| GraphSAGE | $79.89 \pm 0.46$ | $35.20 \pm 0.20$ | $70.20 \pm 0.59$ | $65.77 \pm 0.43$ | $72.02 \pm 0.16$ | $56.65 \pm 0.50$ | $85.19 \pm 0.11$ |
| GAT | $81.68 \pm 0.41$ | $35.74 \pm 0.19$ | $70.53 \pm 0.40$ | $66.03 \pm 0.24$ | $76.06 \pm 0.30$ | $59.01 \pm 0.52$ | $85.70 \pm 0.08$ |
| GT | $80.90 \pm 0.42$ | $34.38 \pm 0.31$ | $67.45 \pm 0.82$ | $64.02 \pm 0.59$ | $75.57 \pm 0.15$ | $58.97 \pm 0.25$ | $85.54 \pm 0.23$ |

Table 6: Experimental results under the TH (temporal high) data split in the transductive setting. The best result and those statistically indistinguishable from it are highlighted in violet. TLE stands for time limit exceeded (24 hours); MLE stands for memory limit exceeded (80 GB VRAM); RTE stands for runtime error in the official code of GFMs.

(a) Results for classification datasets. Accuracy is reported for multiclass classification datasets and Average Precision is reported for binary classification datasets.

| | multiclass classification | | | binary classification | | |
|---|---|---|---|---|---|---|
| | hm-categories | pokec-regions | web-topics | tolokers-2 | artnet-exp | web-fraud |
| best const. pred. | $19.57 \pm 0.00$ | $2.98 \pm 0.00$ | $23.28 \pm 0.00$ | $8.61 \pm 0.00$ | $7.84 \pm 0.00$ | $0.15 \pm 0.00$ |
| ResMLP | $32.44 \pm 0.54$ | $4.18 \pm 0.01$ | $35.49 \pm 0.03$ | $21.72 \pm 6.69$ | $37.48 \pm 0.51$ | $2.83 \pm 0.26$ |
| XGBoost | $31.95 \pm 0.15$ | $4.13 \pm 0.02$ | TLE | $18.37 \pm 6.34$ | $37.48 \pm 0.28$ | TLE |
| LightGBM | $32.08 \pm 0.25$ | $4.13 \pm 0.01$ | TLE | $15.85 \pm 3.89$ | $37.34 \pm 0.18$ | TLE |
| CatBoost | $33.24 \pm 0.16$ | TLE | TLE | $13.87 \pm 1.55$ | $38.56 \pm 0.10$ | TLE |
| ResMLP-NFA | $45.04 \pm 0.26$ | $6.07 \pm 0.07$ | MLE | $38.50 \pm 2.47$ | $36.13 \pm 0.41$ | MLE |
| LightGBM-NFA | $52.45 \pm 0.18$ | $5.54 \pm 0.01$ | TLE | $31.81 \pm 7.51$ | $39.89 \pm 0.37$ | TLE |
| GCN | $56.91 \pm 0.55$ | $11.88 \pm 0.31$ | $38.20 \pm 0.19$ | $46.72 \pm 1.19$ | $40.64 \pm 0.40$ | $4.85 \pm 1.42$ |
| GraphSAGE | $59.62 \pm 0.51$ | $16.60 \pm 0.28$ | $39.00 \pm 0.09$ | $17.05 \pm 7.65$ | $40.50 \pm 0.84$ | $16.01 \pm 2.22$ |
| GAT | $61.28 \pm 0.97$ | $20.43 \pm 0.55$ | $39.24 \pm 0.23$ | $38.59 \pm 6.19$ | $41.85 \pm 0.63$ | $16.50 \pm 1.14$ |
| GT | $63.31 \pm 0.45$ | $25.09 \pm 0.58$ | $39.19 \pm 0.15$ | $34.15 \pm 4.81$ | $40.10 \pm 0.60$ | $11.97 \pm 1.13$ |
| OpenGraph (ICL) | $5.76 \pm 1.03$ | $0.80 \pm 0.45$ | RTE | $9.12 \pm 1.74$ | $16.19 \pm 1.36$ | RTE |
| AnyGraph (ICL) | $9.47 \pm 1.13$ | $9.20 \pm 0.67$ | $11.14 \pm 5.16$ | $13.52 \pm 4.74$ | $11.80 \pm 1.01$ | $0.16 \pm 0.01$ |
| TS-GNN (ICL) | $18.91 \pm 0.32$ | MLE | MLE | $12.59 \pm 1.97$ | $9.46 \pm 1.25$ | MLE |
| GCOPE (FT) | $19.14 \pm 0.58$ | TLE | TLE | $8.46 \pm 1.15$ | $20.83 \pm 1.18$ | TLE |

(b) Results for regression datasets. $R^2$ is reported for all datasets.

| | hm-prices | avazu-ctr | twitch-views | artnet-views |
|---|---|---|---|---|
| best const. pred. | $-2.85 \pm 0.00$ | $0.00 \pm 0.00$ | $-22.31 \pm 0.00$ | $-9.32 \pm 0.00$ |
| ResMLP | $61.64 \pm 0.79$ | $20.35 \pm 1.50$ | $11.91 \pm 8.00$ | $42.15 \pm 0.52$ |
| XGBoost | $63.96 \pm 0.36$ | $25.87 \pm 0.14$ | $-8.84 \pm 0.24$ | $42.45 \pm 0.09$ |
| LightGBM | $63.56 \pm 0.39$ | $26.23 \pm 0.04$ | $-9.60 \pm 0.03$ | $42.87 \pm 0.08$ |
| CatBoost | $62.05 \pm 0.68$ | $25.49 \pm 0.09$ | $-8.76 \pm 0.23$ | $42.83 \pm 0.06$ |
| ResMLP-NFA | $63.70 \pm 1.22$ | $38.00 \pm 0.28$ | $44.38 \pm 1.65$ | $47.17 \pm 0.59$ |
| LightGBM-NFA | $71.36 \pm 0.41$ | $38.69 \pm 0.03$ | $43.60 \pm 0.03$ | $52.42 \pm 0.06$ |
| GCN | $65.20 \pm 0.84$ | $37.49 \pm 0.26$ | $68.17 \pm 0.24$ | $54.44 \pm 0.43$ |
| GraphSAGE | $67.93 \pm 1.24$ | $38.38 \pm 0.39$ | $61.46 \pm 0.68$ | $51.87 \pm 0.48$ |
| GAT | $70.83 \pm 0.99$ | $39.21 \pm 0.17$ | $66.32 \pm 0.59$ | $52.30 \pm 0.34$ |
| GT | $69.70 \pm 0.84$ | $38.27 \pm 0.27$ | $65.83 \pm 0.24$ | $51.67 \pm 0.54$ |

Table 7: Experimental results under the `THI` (temporal high / inductive) setting. The best result and those statistically indistinguishable from it are highlighted in blue. `TLE` stands for time limit exceeded (24 hours); `MLE` stands for memory limit exceeded (80 GB VRAM); `RTE` stands for runtime error in the official code of GFMs.

(a) Results for classification datasets. Accuracy is reported for multiclass classification datasets and Average Precision is reported for binary classification datasets.

| | multiclass classification | | | binary classification | | |
|---|---|---|---|---|---|---|
| | `hm-categories` | `pokec-regions` | `web-topics` | `tolokers-2` | `artnet-exp` | `web-fraud` |
| best const. pred. | $19.57 \pm 0.00$ | $2.98 \pm 0.00$ | $23.28 \pm 0.00$ | $8.61 \pm 0.00$ | $7.84 \pm 0.00$ | $0.15 \pm 0.00$ |
| ResMLP | $32.44 \pm 0.54$ | $4.18 \pm 0.01$ | $35.49 \pm 0.03$ | $21.72 \pm 6.69$ | $37.48 \pm 0.51$ | $2.83 \pm 0.26$ |
| XGBoost | $31.95 \pm 0.15$ | $4.13 \pm 0.02$ | TLE | $18.37 \pm 6.34$ | $37.48 \pm 0.28$ | TLE |
| LightGBM | $32.08 \pm 0.25$ | $4.13 \pm 0.01$ | TLE | $15.85 \pm 3.89$ | $37.34 \pm 0.18$ | TLE |
| CatBoost | $33.24 \pm 0.16$ | TLE | TLE | $13.87 \pm 1.55$ | $38.56 \pm 0.10$ | TLE |
| ResMLP-NFA | $44.99 \pm 0.86$ | $4.95 \pm 0.09$ | MLE | $43.43 \pm 2.04$ | $36.36 \pm 0.49$ | MLE |
| LightGBM-NFA | $47.46 \pm 0.28$ | $4.58 \pm 0.02$ | TLE | $45.45 \pm 0.82$ | $39.91 \pm 0.25$ | TLE |
| GCN | $47.05 \pm 1.69$ | $6.88 \pm 0.51$ | $37.76 \pm 0.06$ | $32.43 \pm 8.03$ | $41.28 \pm 0.28$ | $3.44 \pm 0.33$ |
| GraphSAGE | $48.11 \pm 2.08$ | $8.04 \pm 0.26$ | $38.04 \pm 0.18$ | $30.86 \pm 9.48$ | $40.53 \pm 0.40$ | $13.88 \pm 1.32$ |
| GAT | $59.34 \pm 1.09$ | $13.38 \pm 0.35$ | $38.77 \pm 0.35$ | $24.53 \pm 9.55$ | $41.64 \pm 0.32$ | $11.98 \pm 1.54$ |
| GT | $59.54 \pm 1.59$ | $17.22 \pm 0.42$ | $38.78 \pm 0.08$ | $22.89 \pm 10.40$ | $40.26 \pm 0.82$ | $7.84 \pm 2.35$ |
| OpenGraph (ICL) | $5.76 \pm 1.03$ | $0.80 \pm 0.45$ | RTE | $9.12 \pm 1.74$ | $16.19 \pm 1.36$ | RTE |
| AnyGraph (ICL) | $9.47 \pm 1.13$ | $9.20 \pm 0.67$ | $11.14 \pm 5.16$ | $13.52 \pm 4.74$ | $11.80 \pm 1.01$ | $0.16 \pm 0.01$ |
| TS-GNN (ICL) | $18.91 \pm 0.32$ | MLE | MLE | $12.59 \pm 1.97$ | $9.46 \pm 1.25$ | MLE |
| GCOPE (FT) | $15.69 \pm 3.44$ | TLE | TLE | $10.73 \pm 1.48$ | $19.10 \pm 0.96$ | TLE |

(b) Results for regression datasets. $R^2$ is reported for all datasets.

| | `hm-prices` | `avazu-ctr` | `twitch-views` | `artnet-views` |
|---|---|---|---|---|
| best const. pred. | $-2.85 \pm 0.00$ | $0.00 \pm 0.00$ | $-22.31 \pm 0.00$ | $-9.32 \pm 0.00$ |
| ResMLP | $61.64 \pm 0.79$ | $20.35 \pm 1.50$ | $11.91 \pm 8.00$ | $42.15 \pm 0.52$ |
| XGBoost | $63.96 \pm 0.36$ | $25.87 \pm 0.14$ | $-8.84 \pm 0.24$ | $42.45 \pm 0.09$ |
| LightGBM | $63.56 \pm 0.39$ | $26.23 \pm 0.04$ | $-9.60 \pm 0.03$ | $42.87 \pm 0.08$ |
| CatBoost | $62.05 \pm 0.68$ | $25.49 \pm 0.09$ | $-8.76 \pm 0.23$ | $42.83 \pm 0.06$ |
| ResMLP-NFA | $65.66 \pm 1.04$ | $35.81 \pm 0.56$ | $36.98 \pm 1.44$ | $48.26 \pm 0.82$ |
| LightGBM-NFA | $68.88 \pm 0.11$ | $33.04 \pm 0.17$ | $24.81 \pm 0.11$ | $51.95 \pm 0.05$ |
| GCN | $64.31 \pm 0.82$ | $34.78 \pm 0.48$ | $63.58 \pm 0.54$ | $53.73 \pm 0.47$ |
| GraphSAGE | $65.80 \pm 0.56$ | $36.79 \pm 0.55$ | $56.60 \pm 0.71$ | $53.37 \pm 0.30$ |
| GAT | $69.74 \pm 1.50$ | $37.18 \pm 1.02$ | $61.41 \pm 1.30$ | $52.44 \pm 0.59$ |
| GT | $67.33 \pm 2.05$ | $36.49 \pm 0.83$ | $60.67 \pm 1.02$ | $52.26 \pm 0.48$ |

