# OpenReview forum: "GraphLand: Evaluating Graph Machine Learning Models on Diverse Industrial Data"
_NeurIPS.cc/2025/Datasets_and_Benchmarks_Track — NeurIPS 2025 Datasets and Benchmarks Track poster_

### Official Review · Reviewer_F86w · 2025-06-25

**Rating:** 4
**Confidence:** 4

**Summary:**

This paper introduces GraphLand, a new benchmark suite consisting of 14 large-scale and diverse graph datasets from real-world industrial domains, covering both classification and regression tasks. The authors highlight that existing benchmarks are overly focused on academic citation networks and fail to reflect the structural and feature diversity found in practical applications. GraphLand addresses this gap by including graphs with rich numerical and categorical node features, varied structures, and support for realistic temporal and inductive evaluation settings. Extensive experiments show that attention-based GNNs generally outperform others, while traditional GBDT models with graph-based feature augmentation also perform strongly in regression tasks. In contrast, current general-purpose graph foundation models perform inconsistently and lack robustness across domains. GraphLand provides a more realistic evaluation platform for GML research and points toward important future directions, such as improving model robustness under temporal shifts and cross-domain generalization.

**Dataset Code Accessibility:**

Yes

**Ethical Considerations:**

No, there are no or only very minor ethics concerns

**Final Justification:**

The author's rebuttal has addressed my concerns, so I have increased my score.

**Limitations Weaknesses:**

1. The dataset description is incomplete. While the authors provide the meanings of nodes and edges in the dataset, as well as an overview of the downstream tasks, they do not specify the detailed composition of the node features. Instead, they only mention the dimensionality of the features in general terms. In fact, even when datasets use users as nodes, the specific user features collected can vary significantly. Therefore, for the purpose of dataset release, it is crucial to include a detailed description of the node features.
2. One of the main contributions of GraphLand is to provide a more reliable and fair evaluation for GFMs. However, the paper only includes three GFM models in its experiments. While it is understandable that some GFMs may not have publicly available pre-trained parameters, this should not be a sufficient reason to omit them entirely. Furthermore, since node feature dimensions differ across datasets, it remains unclear how these GFMs are applied to downstream tasks using a unified set of parameters. These implementation details are crucial and should be explicitly stated.
3. Many of the experimental results on GFMs in the paper are limited to RTE and TLE, with only a few meaningful downstream task results reported. This raises concerns about whether the benchmark truly facilitates the development and evaluation of GFMs. If most GFMs struggle to produce usable results under this benchmark, it may suggest that the benchmark itself is not well-aligned with the practical use or capabilities of existing GFMs. This issue deserves further clarification and possibly a reconsideration of the benchmark design.
4. For some datasets such as city-reviews and city-roads, graph-based methods do not show significant improvements over graph-agnostic baselines. This raises the question of whether the graph structure in these datasets is actually useful for the tasks being evaluated. If the graph structure does not contribute meaningfully to performance, it may contradict the second criterion stated for dataset selection: “Datasets should have graph structure that is beneficial for the considered task.” This issue warrants further discussion or justification in the paper.
5. Overall, although the intended target of this benchmark is Graph Foundation Models (GFMs), the paper includes very limited content specifically related to GFMs. This weakens the relevance and impact of the paper for its stated readers.

**Strengths Contributions:**

1. This paper is well-organized and easy to follow.
2. The newly collected datasets in the paper are quite innovative, offering additional perspectives for graph datasets.

---

> ### Author Rebuttal · Authors · 2025-07-31
>
> Thank you for your review. We appreciate that you find our datasets innovative and consider that they offer additional perspectives for the field.
>
> L1. Due to page limits we were not able to provide full description of our datasets in the main text, however, note that Appendix A.1 contains more detailed descriptions including examples of node features for each dataset. While we do not list all the node features, as some of our datasets have several hundred of them, we note that for most datasets feature names can be found as column headers in the corresponding `features.csv` file (and feature types are provided in the corresponding `info.yaml` file). Please let us know if you are interested in the detailed descriptions of some particular features so we can provide them.
>
> L2. Indeed, we were surprised that, despite there being a lot of talk recently about Graph Foundation Models, we were able to find only 3 models that can handle non-textual node features and have open code and weights (and we did a very thorough literature search). If you are aware of other such models, please let us know and we will evaluate them. Unfortunately, we cannot hope to evaluate models for which we have access to neither weights nor training code (especially since their descriptions are often incomplete). As for how these models handle node features of different dimensions in different datasets, we describe it in Appendix B.1 (they typically apply truncated SVD to the feature matrix). However, our results show that these methods are not very effective. As we note in lines 136-142, the same problem of handling diverse features is faced in the field of ML for tabular data, and several successful models have been designed there recently, but those methods have not spread to the graph ML field yet. We hope our benchmark encourages the adoption of such methods to future GFMs.
>
> L3. We would like to clarify that the main aim of our benchmark is to introduce diverse and practically relevant datasets to the field of graph ML. We believe the field should focus more on realistic datasets rather than a few small academic citation networks. We hope our benchmark will encourage such a shift in focus. Our benchmark can be used to evaluate different kinds of models including GFMs. The fact that GNNs can handle our datasets but current GFMs cannot handle some of them is, as we believe, not a limitation of our benchmark, but a limitation of currently available GFMs. We think that the development of GFMs is only in its infancy (as evidenced by a lack of open and generalizable models), and we hope that an introduction of a realistic graph benchmark will steer it towards creating models that can perform well on real-world tasks rather than toy examples, and thus make much more practical impact.
>
> L4. Please note that on all our datasets the best results are achieved by graph-based models, which shows that graph topology is indeed helpful for the considered tasks. In particular, on `city-reviews` and `city-roads`, the best results are achieved by LightGBM-NFA which is a graph-based model due to the use of the NFA module that relies on the graph structure. We verify that graph topology is helpful for every dataset by comparing the results of graph-agnostic ResMLP to the results of graph-aware GNNs and the results of graph-agnostic LightGBM to the results of graph-aware LightGBM-NFA (as we discuss in lines 302-308).
>
> L5. We would like to emphasize that our benchmark is intended for use with *any* graph ML models. Besides several GFMs, we also evaluated GNNs and graph-augmented GBDT models. We mention GFMs in particular because several of them have been proposed recently, but they are often evaluated on datasets with very particular and not realistic characteristics (small size, lack of node features). We hope our benchmark will encourage the development of more practically relevant GFMs.

---

> > ### Comment · Reviewer_F86w · 2025-08-02
> >
> > Thank you for the author's response. I will consider the other reviewers' comments before making my final decision.

---

> > > ### Author Response · Authors · 2025-08-06
> > >
> > > Thank you again for your review and careful consideration of our work! Based on the Author-Reviewer discussion, it seems that our responses have addressed the concerns of the other reviewers. Could you please clarify whether your concerns have also been addressed, or if you have any additional comments or questions? We are open to further discussions.

---

### Official Review · Reviewer_Yois · 2025-06-26

**Ethics Flags:** Data privacy, copyright, and consent
**Rating:** 4
**Confidence:** 4

**Summary:**

This paper proposes a novel node-level graph benchmark comprising diverse real-world datasets from various domains, beyond the widely used citation networks.
The benchmarks include datasets from social networks, web graphs, and road networks, which have diverse structural properties as well as rich node attributes.
The datasets also cover a range of graph sizes for evaluation with different computational resources. However, each of them contains at least 10K nodes to avoid noisy evaluation.
The benchmark also offers different splitting settings, from random splitting to temporal splitting, from transductive settings to inductive settings.

**Additional Feedback:**

(Question) The evaluation on ResMLP: Do you reorder the nodes during evaluation to make sure the order of nodes is not impacting the performance? A graph model should be invariant to the node order, but ResMLP is not, if I understand correctly.

**Dataset Code Accessibility:**

Yes

**Dataset Code Comments:**

The data and code are accessible and reproducible in an executable format.

**Ethical Comments:**

Some datasets might have concerns involving data privacy, copyright, and consent:
1. (web-traffic, web-fraud, and web-topics): (a) potential for identifying specific users by their behavior. (b) the consent from the websites to be tracked.
2.  (artnet-views and artnet-exp): The social network of art creators involving the subscripton as well as the tendency to create explicit art content. This might require the consent from the art creators as well as the social platform. There might be risks of exposing the private information of the art creators, which demands attention from the authors.

**Ethical Considerations:**

Yes, there are ethics concerns that require attention by the authors

**Final Justification:**

The authors provide a proper explanation of my questions.

Considering the rating of other reviewers, I will keep my recommended score.

**Limitations Weaknesses:**

1. The term of ``graph sizes'' shall be utilized carefully: as in graph theory, the size of a graph is referred to the number of edges while the order is the number of vertices.


2. The motivation of the graph foundation models needs more discussion. A large-scale graph datasets does not necessarily equal to a dataset with a large-scale graphs. A dataset with a massive number of graphs, each of which is relatively small, is also a large-scale dataset.
In fact, the latter is more close to the conventional understanding of large-scale datasets in other domains, while the former is more related to the long-context learning in language tasks and high-resolution images learning.
However, they are highly related and tied to each other. A more detailed discussion can make the motivation stronger.
The discussion might include the following points:
   - In each dataset: one graph with a large number of nodes v.s. a large number of graphs with a moderate number of nodes
   - Inductive setting:  unseen nodes in the same graph v.s. completely unseen graphs (node-level tasks can also have the multiple-graph inductive setting, see Pattern/Cluster from Dwivedi et al. (2023))

3. There are no discussion on why focusing on node-level tasks. There are also graph-level and edge-level tasks, which are also important and useful for graph learning. (I don't think the focus on node-level tasks is bad, but it is worth discussing why node-level tasks are chosen and what the advantages are.)



-- Dwivedi, Vijay Prakash, et al. "Benchmarking graph neural networks." Journal of Machine Learning Research 24.43 (2023): 1-48.

**Strengths Contributions:**

1. The proposed benchmark addresses several limitations of existing node-level graph benchmarks, such as limited domain diversity and potentially noisy evaluations.
2. It ensures that the provided graph structure is not noisy and beneficial to the learning tasks.
3. Solid data splitting strategies and experimental settings are provided with careful discussion.
4. Empirical experiments demonstrate that the benchmark effectively distinguishes the ability of graph models to leverage structural information in the graph learning tasks.
This is especially important for evaluating models with strong attribute encoders (e.g., LLMs), as in some datasets, node attributes may already encode structural information—allowing models with limited structural modeling capacity to perform well if the attribute encoder is strong.

---

> ### Author Rebuttal · Authors · 2025-07-31
>
> Thank you for your review and positive feedback. We address your concerns below.
>
> L1. Thank you, we will clarify this in the updated text.
>
> L2. Indeed, in GML, there are currently two most popular dataset types: one relatively large graph with node-level or edge-level tasks (e.g., a social network) and many relatively small graphs with typically graph-level tasks (e.g., molecules). Both settings have important practical applications. In our work, we focus on the first setting. In this setup, inductiveness typically refers to new nodes and edges appearing in an evolving graph. Although having a completely new graph for some node-level task is possible, such a scenario may be not so common in practice, and the corresponding datasets may be very different from those introduced in our work. For instance, regarding the mentioned Pattern and Cluster datasets, they represent pattern recognition and clustering tasks on synthetically generated graphs from stochastic block models. In contrast, our work is focused on the node property prediction tasks on real-world graphs from diverse industrial applications. We will clarify this in the updated text, thank you for this comment.
>
> L3. Indeed, node-level, graph-level, and edge-level tasks are all very important. They appear in very diverse applications, thus covering all of them simultaneously is not easy. In our work, we choose to focus on node-level tasks as we believe they have the most serious lack of realistic and practically relevant benchmarks (and many models are only evaluated on a few popular academic citation networks, which represent a very narrow domain). In contrast, for graph-level tasks, there exist several large-scale molecular datasets. We plan to extend our discussion of node-level problems in the revised text.
>
> > Some datasets might have concerns involving data privacy, copyright, and consent...
>
> Thank you for bringing up this issue. We would like to clarify that all the data for our new datasets was collected in accordance with the terms of use of the corresponding platforms and our datasets do not contain any personally identifiable information. Further, our data collection process has been rigorously evaluated by our company’s legal team and it has been confirmed that the datasets can be legally provided to the open community. We hope these clarifications address your concern.
>
> > The evaluation on ResMLP: Do you reorder the nodes during evaluation to make sure the order of nodes is not impacting the performance? A graph model should be invariant to the node order, but ResMLP is not, if I understand correctly.
>
> Our ResMLP refers to a model that treats all nodes as independent data samples and does not consider the graph topology at all. It is trained on batches of independent samples and does not depend on the order of these samples. We will clarify this in the updated text. We use such a model to verify that GNNs perform better on our datasets than a graph-agnostic model and thus the graph topology is actually beneficial for the considered tasks.

---

> > ### Comment · Reviewer_Yois · 2025-08-04
> >
> > Thank you for the detailed responses.
> >
> > Overall, I believe the work can contribute to enriching the benchmarks for node-level tasks.
> > To be honest, at this stage, it falls somewhere between an accept and a borderline accept for me.
> >
> > I will maintain my current score (borderline accept) for now, but I will consider increasing it after the discussion with other reviewers.
> >
> >
> > > Minor question:
> > How do existing models behave in terms of overfitting on the proposed benchmarks?
> >
> > I ask this because many prior benchmarks fail to reliably measure model capacity due to overfitting issues—often stemming from limited label availability or label imbalance. In such cases, more expressive models may underperform compared to weaker ones, not due to lack of capability, but due to overfitting to the training data.
> >
> > Could the authors comment on how severe overfitting is on their proposed benchmarks? Specifically:
> > - How do training and validation metrics compare to test performance?

---

> > ### Author Response · Authors · 2025-08-04
> >
> > Thank you for this question! We would like to note that even our smaller datasets are several times larger than the `cora` and `citeseer` datasets often used for model evaluation in graph ML, and thus provide more reliable model performance estimates. In particular, we can see that more complex GAT and especially GT models often perform significantly better than simpler GCN and GraphSAGE models. Further, our data splits with different train set proportions allow investigating the difference in model performance depending on the number of labeled nodes. In practice, we observe little to no difference between validation and test set performance under random data splits for all our datasets. However, the situation changes when the temporal split is used. With the temporal TH data split, models often exhibit significantly worse performance on test data than on validation data. This difference often becomes even more drastic when the inductive setting THI is used and thus test notes are not seen during validation (and validation nodes are not seen during training). In this setting, models tend to exhibit very stark differences between train/validation/test set performance. For example, on the `tolokers-2` dataset GT model achieves average precision of 33.7 on the validation set, but only 22.9 on the test set. This demonstrates that GNNs can be very vulnerable to temporal distribution shifts and structural changes in dynamically evolving graphs, a problem which was mostly overlooked in the literature previously despite such shifts being very common in industrial applications. We hope our benchmark encourages the development of graph ML models more resilient to distribution shifts.

---

> > > ### Comment · Reviewer_Yois · 2025-08-06
> > >
> > > Thank you for your explanation.
> > >
> > > The temporal TH data split is quite interesting and deserves further discussion.
> > >
> > > For example, it may be worthwhile to consider whether temporal distribution shifts require explicit modeling beyond what is offered by standard GNN learning pipelines.
> > >
> > > Best of luck with your submission.

---

### Official Review · Reviewer_2dkq · 2025-06-27

**Ethics Flags:** Data privacy, copyright, and consent
**Rating:** 4
**Confidence:** 3

**Summary:**

The authors present fourteen node-level graph learning tasks. The datasets span web graphs (nodes = websites), various social networks (nodes = users), road networks (nodes = road segments), and purchase networks (nodes = products).
Several of the datasets contain temporal information; the authors provide up to three train/validation/test splits with a temporal split that turns out to be very challenging for existing graph learning models.

**Additional Feedback:**

I am happy to raise my score if the ethics review board considers the flagged issues minor. Generally, the datasets would be a nice extension of the existing benchmark landscape. In particular, the realistic time splits could be a novel challenging task.

**Dataset Code Accessibility:**

Yes

**Dataset Code Comments:**

The authors provide their code via anonymized github and the datasets via anonymous kaggle.

**Ethical Comments:**

Given that several presented learning tasks are to predict properties of users of certain platforms, I believe that the authors (a) do not provide any information about consent of the users of these platforms to have their data included in such a dataset and (b) do not address the potential negative impact of training models to predict e.g. number of clicks/views or fraud. Given that creators or workers (e.g. on artnet-, tolokers-, twitch) may base a potentially large part of their income on such platforms, well (or poor) performing models for the proposed tasks may impact them negatively.

The paper also does not discuss anonymization of the data. How likely is a deanonymization of the nodes in these datasets.

**Ethical Considerations:**

Yes, there are significant ethics concerns that require review by an ethics expert

**Final Justification:**

The authors have provided clarifications regarding the ethical questions I have raised. To me, it sounds sufficient. However, the decision whether to accept this paper should depend on whether the ethics reviewer decides that the answers provided by the authors are sufficient. I don't consider myself competent to make this decision.

Regarding the technical content of the paper, I do believe that the contribution is relevant to the graph learning community. However, the description provided in the discussion is not detailed enough to be a proper reference if somebody is really interested in understanding the provenance of the data. I would like to recommend acceptance, given that the authors extend the description dramatically in the final paper.

**Limitations Weaknesses:**

Several of the presented datasets are not described in detail. In particular web-*, artnet-*, and city-* don't provide any reference to a previous source or sufficient description in the paper (or appendix) of the sourcing process. Did the authors collect the data themselves? Over which time window? What do the node features encode? Is the data collection from the sources acceptable under the Terms of Use of the respective platforms? How were these datasets collected? Were all users of these platforms scraped, or is it a sample of all available data? If so, how was the sample constructed?

**Strengths Contributions:**

The paper is well structured and easy to follow. It motivates the proposed datasets with a relative lack of node level benchmark datasets beyond academic citation networks.
The authors prepare up to three dataset splits (10/10/80 transductive), (50/25/25 transductive), and (50/25/25 inductive, split at a certain time stamp). This allows to use these datasets for standartized comparison in a wide range of settings.

---

> ### Author Rebuttal · Authors · 2025-07-31
>
> Thank you for your review. We are glad that you find our datasets a useful extension of the existing benchmarks. We address your concerns below.
>
> > Several of the presented datasets are not described in detail. In particular web-, artnet-, and city-* don't provide any reference to a previous source or sufficient description in the paper (or appendix) of the sourcing process. Did the authors collect the data themselves? Over which time window? What do the node features encode? Is the data collection from the sources acceptable under the Terms of Use of the respective platforms? How were these datasets collected? Were all users of these platforms scraped, or is it a sample of all available data? If so, how was the sample constructed?
>
> Thank you for bringing up this issue. We would like to clarify that the only reason we did not provide a detailed description of data sources for the `web-*`, `artnet-*`, and `city-*` datasets is to preserve anonymity of our submission, as disclosing the exact services from which the data was collected will also disclose our affiliation, since these services are run by the company we are affiliated with. We believe that our decision does not create any problems for proper review of our work as our datasets are fully available to the reviewers. Upon the acceptance, we will be able to provide more details regarding the sources of data for the mentioned datasets. Further, we would like to emphasize that all the data was collected in accordance with the terms of use of the corresponding platforms. In fact, our company’s legal team rigorously evaluated our data collection process and confirmed that the datasets can be legally provided to the open community. In particular, our datasets do not contain any personally identifiable information. Data collection periods for our datasets cover different time windows depending on the dataset: two months for `city-reviews`, almost a year for `artnet-*` and slightly more than a year for `web-*`. We hope these clarifications address your concern.
>
> > Given that several presented learning tasks are to predict properties of users of certain platforms, I believe that the authors (a) do not provide any information about consent of the users of these platforms to have their data included in such a dataset and (b) do not address the potential negative impact of training models to predict e.g. number of clicks/views or fraud. Given that creators or workers (e.g. on artnet-, tolokers-, twitch) may base a potentially large part of their income on such platforms, well (or poor) performing models for the proposed tasks may impact them negatively.
>
> As mentioned above, our data was collected in accordance with the terms of use of the corresponding platforms and does not contain any personally identifiable information. As for the impact of training models for fraud detection and engagement prediction tasks, we would like to note that both tasks are standard for modern technological companies and internet services. While models poorly performing in these tasks can indeed lead to negative societal consequences, we believe that the existence of open anonymized high-quality datasets for such tasks actually leads to the development of better performing models and thus to positive consequences for society. That is why we tried to include the datasets covering these important applications into our benchmark. We plan to extend the section describing the potential societal impacts of our work, and we will be glad to discuss this question with the ethics experts to ensure that our benchmarking efforts responsibly address any ethical concerns.

---

> > ### Comment · Reviewer_2dkq · 2025-08-04
> >
> > Thank you for these clarifications. In light of the statements regarding the accordance with the terms of use, and the ethics review that identifies this (and data provenance) as the main issues, I'd be happy to raise my score. Nevertheless, I am a bit reluctant to do so. I'd like to point out that the D&B track allows single blind submissions to avoid situations like the current one, where anonymity of the submission would require a relevant part of the paper to be left out.
> >
> > Could the authors present a (draft of) the description here?

---

> > > ### Author Response · Authors · 2025-08-04
> > >
> > > Thank you for your reply. To follow an established review process, we opted for a double blind submission and we did not expect it to raise any issues. However, since you are still interested in the description of our data sources we can disclose that the `artnet-*` datasets are based on data from Shedevrum art creation platform, `city-roads-*` datasets are based on traffic data from the Yandex Maps service, `city-reviews` dataset is constructed based on reviews from the Yandex Maps places and organizations reviews service, and `web-*` datasets represent a random sample of the web-graph collected with Yandex’s internet search engine. We would like to emphasize once again that all our data was collected in accordance with the terms of use of the corresponding services and our datasets were reviewed by the legal teams of these services which confirmed that these datasets can be released publicly under the Apache 2.0 licence. We have provided this information and more details in a private response to the ethics review, which we believe settles all concerns regarding potential ethical issues. Please let us know if there are any specific details about the dataset creation or properties you would like us to specify or if you have any other concerns. Thank you once again for your reviewing effort.

---

> > > > ### Comment · Reviewer_2dkq · 2025-08-05
> > > >
> > > > Thank you for the additional details. For me, in this context, this is sufficient. I have increased my score.  However, I recommend that for the final version of this paper, further details be provided, e.g. time windows of data collection, how the 'random sample' of the web graph was selected, etc.

---

> > > > > ### Author Response · Authors · 2025-08-06
> > > > >
> > > > > Thanks for your support and positive feedback! Following your suggestion, we will include additional details about the process of data collection in the revised text.

---

### Official Review · Reviewer_wAk3 · 2025-06-28

**Rating:** 4
**Confidence:** 4

**Summary:**

This paper introduces GraphLand, a large-scale and diverse benchmark for GML, covering 14 graph datasets with node property prediction tasks beyond standard citation networks. The authors emphasize dataset scale (each with at least 10,000 nodes), structural diversity, and provide multiple data splits, including temporal and inductive settings. The benchmark aims to enable more realistic and comprehensive evaluations of GNNs and GFMs in industrial-like settings.
Overall, the paper is clearly motivated, and I appreciate the effort in addressing the current limitations of small-scale, narrow-domain GML benchmarks.

**Additional Feedback:**

See Weaknesses.
Overall, this is a very good and timely piece of work, addressing a real need in the community for larger, diverse, and realistic GML benchmarks. With improvements to benchmarking rigor and clearer positioning of industrially relevant tasks, this benchmark could become highly impactful.

**Dataset Code Accessibility:**

Yes

**Ethical Considerations:**

No, there are no or only very minor ethics concerns

**Final Justification:**

The concerns that initially led to a borderline reject rating have been largely addressed.

**Limitations Weaknesses:**

While I find this paper promising, I see several areas where improvements could further strengthen its contribution:

1. The authors note that each dataset has at least 10,000 nodes, which is excellent for reducing evaluation noise. However, Table 1 (Characteristics of the proposed GraphLand datasets), currently in the appendix, would be highly beneficial if moved into the main paper to give readers a quick overview of dataset sizes and structures.

2. While the datasets are diverse, several prediction tasks may lack practical industrial relevance:

   - web-topics (predicting website topics), hm-categories (predicting product categories) and pokec-regions (predicting user regions) do not reflect clear industrial demand, as such metadata is typically directly available.

   - twitch-views involves view counts that heavily depend on content quality and platform exposure, which are not strongly related to the user subscription graph.

   - artnet-exp (explicit content prediction) may depend on uploaded content rather than user connections.

3. Recent work [1,2,3,4] suggests that classic GNNs (GCN, GraphSAGE, GAT) have been underestimated in general graph tasks and can achieve SOTA performance with hyperparameter tuning. To better reflect their true potential, it would be helpful to consider the impact of hyperparameters:

   - The current setup uses 3 layers, but classic GNNs can be evaluated with up to 15 layers using residual connections to test deeper architectures.

   - Dropout rates are limited to {0, 0.1, 0.2}; testing higher dropout rates (e.g., 0.5, 0.7) could improve robustness.

   - Both Batch Normalization (BN) and Layer Normalization (LN) should be systematically evaluated.

   - The addition of FFN could further enhance performance.

   These adjustments would ensure a fairer comparison between GNNs, GTs, and GFMs, providing a more accurate benchmarking landscape.

4. In Appendix line 186, the authors describe GT as a local graph transformer where each node attends only to its neighbors. I would suggest clarifying that this architecture may not fully align with the standard definition of Graph Transformer, which typically involves global attention across nodes, to avoid potential confusion.

[1] A Critical Look at the Evaluation of GNNs Under Heterophily: Are We Really Making Progress? ICLR 2023.

[2] Where Did the Gap Go? Reassessing the Long-Range Graph Benchmark, LoG 2023.

[3] Classic GNNs are Strong Baselines: Reassessing GNNs for Node Classification, NeurIPS 2024.

[4] Can Classic GNNs Be Strong Baselines for Graph-level Tasks? Simple Architectures Meet Excellence, ICML 2025.

**Strengths Contributions:**

1. The paper addresses a clear gap in current GML benchmarks, which often rely on citation networks.
2. The authors have curated 14 datasets across various domains (web graphs, road networks, social networks, e-commerce graphs) with a variety of tasks (regression and classification). The diversity in graph structures and node features is commendable.
3. Including temporal and inductive splits allows the community to test models under more realistic settings.
4. The paper is clearly motivated.

---

> ### Author Rebuttal · Authors · 2025-07-31
>
> Thank you for your review. We appreciate that you find our work timely and addressing a real need in the community and that you consider our benchmark to be potentially highly impactful.
>
> L1. Thank you for your suggestion. Due to the page limits, we were not able to fit all the important descriptions in the main text. We will try to restructure the text to fit the table with dataset properties in Section 3.
>
> L2. While some datasets have more directly applied tasks than others, we believe that all our tasks have practical relevance and can appear in industrial settings under certain circumstances. For example, as for the `pokec-regions` dataset, many social network users can choose not to fill the region field, thus it might be useful to predict it for use in applications such as more relevant advertising. As for the `web-topics` dataset, the information about the topic to which a website is dedicated is typically not directly available and has to be predicted, as there is no standard field for such information on web pages. We agree that for the `twitch-views` and `artnet-*` datasets the users’ content is an important source of information. However, as our experimental results show, the graph structure also provides a lot of information, allowing GNNs to achieve very strong performance on the task (it is expected, since network science shows that popular users can be identified from just the network structure, and users with similar interests often tend to form communities in the network). Thus, graph-based models can be used as auxiliary models for the tasks even if user-generated content is also available. Further, some users may choose to hide their created content from non-subscribed users, making the graph structure the only available source of information. Thus, we believe our datasets cover much more practically relevant scenarios than the classic academic citation network datasets.
>
> L3. Thank you for bringing up these works; we are actually big fans of this direction of research. In fact, our classic GNN implementations closely follow the implementations from [1], and we also took inspiration from the codebase of [3]. Thus, our models include FFN modules between message-passing layers, as well as skip connections and layer normalization, as we mention in lines 296-298. These modifications significantly improve the performance of our models. As for the hyperparameter grids, they are somewhat narrow due to computational resource limitations and some of our datasets being quite large and dense. However, we ran preliminary experiments with other hyperparameter configurations and found that the ranges we used for the final grid search tend to lead to the best results in most settings. Indeed, classic GNNs with our hyperparameter search settings achieve the best results on most of our datasets. During the rebuttal period, we were able to run extended hyperparameter sweeps for most datasets, and we report the results in the table below. Here GCN, GraphSAGE, GAT, and GT are our original results reported in the current version of the paper; “extended” indicates an extended hyperparameter sweep with up to 15 layers and dropout values up to 0.7; and “batchnorm” indicates models for which layernorm has been replaced with batchnorm. It can be seen that these changes do not lead to statistically significant improvements, supporting that our original grid search is adequate for our datasets.
>
> |                     | city-roads-M   | city-roads-L   | city-reviews   | artnet-views   | artnet-exp   | twitch-views   | tolokers-2   | avazu-ctr    |
> |:--------------------|:---------------|:---------------|:---------------|:---------------|:-------------|:---------------|:-------------|:-------------|
> | GCN                 | 59.05 ± 0.16   | 53.26 ± 0.14   | 77.15 ± 0.28   | 55.99 ± 0.26   | 43.09 ± 0.38 | 75.55 ± 0.05   | 51.32 ± 0.96 | 30.47 ± 0.27 |
> | GCN-extended        | 59.01 ± 0.14   | 53.10 ± 0.21   | 77.21 ± 0.30   | 56.18 ± 0.21   | 43.10 ± 0.25 | 75.51 ± 0.08   | 52.45 ± 1.02 | 30.31 ± 0.41 |
> | GCN-batchnorm       | 58.97 ± 0.10   | 53.04 ± 0.12   | 76.97 ± 0.32   | 55.91 ± 0.32   | 43.02 ± 0.39 | 75.40 ± 0.16   | 51.30 ± 1.28 | 30.28 ± 0.53 |
> | GraphSAGE           | 57.51 ± 0.53   | 52.43 ± 0.25   | 77.82 ± 0.13   | 49.79 ± 0.51   | 42.65 ± 0.59 | 66.87 ± 0.11   | 53.73 ± 0.53 | 31.84 ± 0.24 |
> | GraphSAGE-extended  | 57.62 ± 0.61   | 52.40 ± 0.20   | 77.90 ± 0.16   | 49.90 ± 0.40   | 42.50 ± 0.70 | 67.05 ± 0.18   | 53.70 ± 0.39 | 31.70 ± 0.32 |
> | GraphSAGE-batchnorm | 57.38 ± 0.74   | 52.29 ± 0.32   | 77.58 ± 0.20   | 49.72 ± 0.62   | 42.32 ± 0.76 | 67.01 ± 0.20   | 53.41 ± 0.71 | 31.45 ± 0.57 |
> | GAT                 | 59.11 ± 0.20   | 53.43 ± 0.20   | 77.67 ± 0.13   | 53.36 ± 0.78   | 46.62 ± 0.32 | 72.93 ± 0.17   | 53.78 ± 1.34 | 33.20 ± 0.20 |
> | GAT-extended        | 58.94 ± 0.26   | 53.36 ± 0.18   | 77.75 ± 0.20   | 53.72 ± 0.80   | 46.58 ± 0.38 | 72.87 ± 0.20   | 53.97 ± 1.51 | 33.18 ± 0.31 |
> | GAT-batchnorm       | 58.81 ± 0.30   | 53.05 ± 0.26   | 77.53 ± 0.21   | 53.25 ± 0.89   | 46.40 ± 0.43 | 72.80 ± 0.29   | 53.35 ± 1.71 | 33.02 ± 0.45 |
> | GT                  | 58.05 ± 0.58   | 53.38 ± 0.12   | 76.97 ± 0.21   | 54.23 ± 0.22   | 45.16 ± 0.46 | 72.19 ± 0.14   | 54.50 ± 1.20 | 30.87 ± 0.47 |
> | GT-extended         | 58.04 ± 0.54   | 53.36 ± 0.41   | 76.98 ± 0.18   | 54.29 ± 0.25   | 45.18 ± 0.41 | 72.08 ± 0.19   | 54.37 ± 1.12 | 31.07 ± 0.50 |
> | GT-batchnorm        | 57.89 ± 0.61   | 53.01 ± 0.48   | 76.79 ± 0.24   | 53.97 ± 0.47   | 44.94 ± 0.51 | 71.94 ± 0.31   | 54.02 ± 1.39 | 30.79 ± 0.63 |
>
> L4. Thank you, we will move this important clarification to the main text.

---

> > ### Comment · Reviewer_wAk3 · 2025-08-05
> >
> > Thank you for the detailed response. I appreciate the additional clarifications and experiments. I have accordingly raised my score.

---

### Note · Authors · 2025-08-12

We thank all the reviewers for the helpful discussion. We are glad that reviewers noted that our benchmark “aims to enable more realistic and comprehensive evaluations of GNNs and GFMs in industrial-like settings”, “addresses a clear gap in current GML benchmarks” and “is a very good and timely piece of work, addressing a real need in the community for larger, diverse, and realistic GML benchmarks” (reviewer wAk3), “would be a nice extension of the existing benchmark landscape” (reviewer 2dkq), “addresses several limitations of existing node-level graph benchmarks, such as limited domain diversity” (reviewer Yois), and that our datasets “are quite innovative, offering additional perspectives for graph datasets” (reviewer F86w).

The reviewers also appreciated our introduction of challenging temporal splits and inductive settings: “including temporal and inductive splits allows the community to test models under more realistic settings” (reviewer wAk3), “the realistic time splits could be a novel challenging task” (reviewer 2dkq), “solid data splitting strategies and experimental settings are provided with careful discussion” (reviewer Yois).

We believe that, given the abundance of graph-structured data in impactful real-world applications, it is time for the graph ML community to move beyond benchmarking node property prediction models mostly on academic citation networks and evaluate these models on more complex and meaningful tasks. Our benchmark offers a set of diverse and realistic industrial datasets that can support the development of the graph ML field in this direction.

---

### Decision · Program_Chairs · 2025-09-18

**Decision:**

Accept (poster)

**Comment:**

This paper presents an extensive collection of large scale graph data from real-world setting. Reviewers have all acknowledged the importance and potential significance of this resource, closing an important gap for the relevant research community. Some concerns were raised regarding the benchmarking strategy (for example, hyperparameter tuning), but these were mostly addressed in discussion. The more significant concerns raised, through, were regarding the lack of details in the description of the data, which would limit the usability and interpretability on one hand, but also raised ethical questions regarding the source of the data. These were addressed in discussion, and it seems part of the ambiguity here was due to misunderstanding on the authors' part of the single-blind review procedure used in this track (as opposed to the double-blind review used in the main track), resulting in deliberate obfuscation for anonymity's sake. While this is reasonably an innocent misunderstanding, it does put the paper in a borderline state (indeed, while all reviewers converged on a positive score - they all marked it as borderline).

In my opinion, the initial draft submitted to the conference is quite far from the acceptance threshold, but the explanations and additional details provided in rebuttal and discussion are convincing. They do, however, need to be incorporated thoroughly into the manuscript itself. If it were possible to have a resubmitted version go through a second round of reviews, that would preferably be my recommendation, but unfortunately this option does not exist within the framework of this track and conference. As a result, within the existing framework, this work has been the subject of a discussion with the senior area chair to weigh the scope of required revision relative to the expected impact of the paper.

Ultimately, we have concluded that the potential impact here is quite high, and outweighs the magnitude of revisions needed. This is a collection of 14 datasets that span a wide variety of applications and tasks, with a consistent set of criteria (or desiderata) applied to decide the composition of data included here. This will be appealing to many researchers working on GML, so I would expect many to be attracted to this work and try to leverage these data towards development, benchmarking, and evaluation, as long as the additional details provided by the authors during the rebuttal and discussion periods is appropriately integrated in the published manuscript to support appropriate utilization of provided resources (and alleviate ethical concerns). Therefore, following this discussion and in agreement with the senior area chair, I recommend acceptance of this submission, conditional on a thorough revision by the authors to address the feedback from the reviewers and incorporate the additional information provided in the discussion and any further details indicated throughout the reviews and discussion.